# Improved Learning Rates of a Functional Lasso-type SVM with Sparse Multi-Kernel Representation

**Shaogao Lv[1], Junhui Wang[4], Jiankun Liu[5], Yong Liu[2,3*]**
[1]Nanjing Audit University
[2]Gaoling School of Artificial Intelligence, Renmin University of China, Beijing, China
[3]Beijing Key Laboratory of Big Data Management and Analysis Methods, Beijing, China
[4]City University of Hong Kong
[5]Institute of Information Engineering, Chinese Academy of Sciences
`lvsg716@nau.edu.cn, liuyonggsai@ruc.edu.cn`

## Abstract

In this paper, we provide theoretical results of estimation bounds and excess risk upper bounds for support vector machine (SVM) with sparse multi-kernel representation. These convergence rates for multi-kernel SVM are established by analyzing a Lasso-type regularized learning scheme within composite multi-kernel spaces. It is shown that the oracle rates of convergence of classifiers depend on the complexity of multi-kernels, the sparsity, a Bernstein condition and the sample size, which significantly improve on previous results even for the additive or linear cases. In summary, this paper not only provides unified theoretical results for multi-kernel SVMs, but also enriches the literature on high-dimensional nonparametric classification.

## 1 Introduction

SVM was first introduced in [45] and has became one of the most popular machine learning algorithms in the past two decades. The standard SVM classification consists of two main ingredients, namely the hinge loss and the kernel embedding. The hinge loss is used to model the learning target and can often generate sparse solutions [38], while the kernel embedding is used to model nonlinear relationship between input features and response [39]. In [38], they provide a detailed overview of SVMs and related learning theory. More work on theoretical perspective of SVM have also been developed in recent years, such as [12, 53, 36, 33, 30, 24], among many others.

It is known that the performance of kernel machines largely depends on the data representation via the choice of kernel function [34, 16, 19, 28, 26, 48, 49]. Towards this direction, many approaches have been proposed for kernel selection under different frameworks. For example, Micchelli et al.[34] attempt to find an optimal kernel from a prescribed convex set of basis kernels; Wu et al.[46] optimize the scale parameter among various Gaussian kernels; and Ong et al.[35] study hyperkernels on the space of kernels and alternative approaches, including kernels selection by DC programming and semi-infinite programming. In particular, the seminal work of [22] proposes the so-called multiple kernel learning (MKL) method, learning SVM and a linear combination of kernels simultaneously, which has received a lot of attention in recent years [3, 50, 20, 19, 16, 23, 17, 27, 25, 29, 51]. To be precise, given a finite (possibly large) dictionary of symmetric positive semidefinite kernels $\{K_m : m = 1, 2, ..., M\}$, one can try to find an 'ideal' kernel $K$ as a convex combination of the kernels from a predefined family of potential candidates: $K \in \mathcal{K} := \left\{ \sum_{m=1}^{M} \theta_m K_m : \sum_{m=1}^{M} \theta_m = 1, \theta_m \geq 0 \right\}$. This combination of multiple kernels corresponds to a

---

*Corresponding Author

35th Conference on Neural Information Processing Systems (NeurIPS 2021).

multi-kernel hypothesis space for learning:

$$\mathcal{H}^M := \left\{ \sum_{m=1}^{M} f_m(x) : \ f_m \in \mathcal{H}_{K_m}, \ x \in \mathcal{X} \right\},$$

where $\mathcal{H}_{K_m}$ is a reproducing kernel Hilbert space (RKHS) induced by the kernel $K_m$, as defined in Section 2. Given the learning rule, $\theta_m$'s also need to be estimated automatically from the training data.

Besides flexibility enhancement, other justifications of MKL have also been proposed; such as each kernel function corresponds to different information source (e.g. text, image, gene), or classifiers from different spaces (corresponding to different kernels) are averaged to achieve better classification accuracy [8]. Under many of these situations, the number of candidate kernels $M$ is often very large, even larger than the sample size $n$ in some extreme cases. Interestingly, when one-dimensional linear (or additive) kernels are used and $M$ is the dimensionality of covariates with $M \geq n$, MKL reduces to high-dimensional linear (or additive) models, which have been widely studied in the statistics and machine learning literature [15, 6, 32].

Given the large number of kernels, it is common that not all the kernels are significantly relevant to the response. To avoid impairing generalization performance [54] and enhance interpretability, many redundant kernels should be removed accordingly. In view of generalization ability, interpretability and computational feasibility, a more parsimonious and flexible algorithm for MKL is to generate a sparse linear combination of kernels based on the training data, namely *sparse MKL*. From a statistical point of view, sparse MKL can be interpreted as a model selection task, and can also be viewed as an extension of sparse linear models or sparse additive models. In other words, sparse MKL provides an appropriate route to tackle the kernel learning issues in machine learning and the high-dimensional issues in statistics.

In literature, many regularized learning algorithms based on different regularization terms have been investigated in multi-kernel regression [3, 20, 18, 37, 41]. Particularly, in [37, 41], they established the optimal rates of convergence for the least square approaches in a related setting. In multi-kernel classification, Christmann and Hable [12] gave consistency and robustness property of non-sparse additive SVM when $M$ is fixed; In [36, 42], they provided upper bounds on the estimation error and the excess risk in high-dimensional linear case or with finite number of basis functions. In summary, all the aforementioned work either focus on the regression problem with the quadratic losses, or only consider the parametric case for high-dimensional SVM.

The main focus of this paper is on a more challenging case with sparse multi-kernel approximation for SVM. That is, while the total number of kernels may be larger than the sample size, only a small number of kernels are needed to represent/approximate the target function, so that such a learning problem in the sense of approximation is sparse. Let $X$ be a random variable of the input space $\mathcal{X}$, and $Y \in \mathcal{Y} = \{-1, +1\}$ be the response. Define the theoretical oracle predictor of the classical SVM within $\mathcal{H}^M$ as

$$f^* = \arg \min_{f \in \mathcal{H}^M} \mathcal{E}(f), \ \text{with} \ \mathcal{E}(f) := \mathbb{E}[\phi(Yf(X))]. \tag{1}$$

Here $\phi(t) := (1-t)_+$ is the hinge loss and the expectation is taken with respect to the joint distribution $\rho$ defined on $\mathcal{X} \times \mathcal{Y}$. In the sparse setting, one often assumes that $f^*$ has an additive and sparse representation within $\mathcal{H}^M$, namely, $f^* = \sum_{m \in S} f_m^*$ with an unknown subset $S \subset \{1, 2, ..., M\}$ and $f_m^* \in \mathcal{H}_m$ with $\|f_m^*\|_{K_m} \leq 1$ for all $m \in S$. Of primary interest is the case with $s := |S| \ll M$. Note that $f^*$ here is an optimal estimator within $\mathcal{H}^M$, as opposed to the classical Bayes decision rule defined over all the measurable functions. Actually, the minimizer of the expected risk corresponding to the hinge loss is not continuous against the conditional probability [5].

The main contribution of this paper is to develop a set of theoretical results on multi-kernel SVM and the rates of convergence for nonparametric classification within the multi-kernel setting, including high-dimensional linear or additive classification as special cases. This paper provides refined error bounds for the excess risk, defined as $\mathcal{E}(\hat{f}) - \mathcal{E}(f^*)$, of the proposed estimator $\hat{f}$, as well as its estimation error $\|\hat{f} - f^*\|_2$. The proposed estimator in (5) is based on a regularization of the hinge loss with two fold $\ell_1$ penalty terms, where one is used to control nonparametric sparsity and the other is to control functional smoothness. Particularly, Corollary 1 shows that with high probability, the

excess risk of the proposed estimator is upper bounded by

$$\frac{s}{\Gamma^2(S, \rho_X)} O\Big(n^{-\frac{1}{1+\tau}} + \frac{\log M}{n}\Big)^{\frac{\kappa}{2\kappa - 1}}$$

under some regularity conditions, where $\tau$ corresponds to the spectral decay of each RKHS and is used to characterize functional complexity of kernels, $\kappa$ is called the Bernstein parameter that reflects the low-density level of decision boundary, and $\Gamma(S, \rho_X)$ is used to characterize correlation structure among multiple RKHS's. The same rate also holds true for $\|\hat{f} - f^*\|_2^{2\kappa}$ following the Bernstein condition and the established results on the excess risk.

Furthermore, we establish the oracle rate of the relative misclassification error of $\hat{f}$ in Theorem 2, which consists of the same rate as the excess risk of the estimator plus some additional sparse approximation error. By using this specific form of RKHS's, we can incorporate many existing results as special cases of MKL for SVMs, and more importantly, several existing results can be improved by our derived ones. The detailed comparison between our general results with existing results is conducted in Section 3.3.

**Notations**. Define the $L_2$ norm of a function $f$ by $\|f\|_2^2 = \int_X |f(x)|^2 d\rho_X(x)$ with marginal distribution $\rho_X$. Given sample points $\{x_i\}_{i=1}^n$, $\|f_m\|_n^2 := \frac{1}{n} \sum_{i=1}^n f_m(x_i)^2$ and $\|f_m\|_n$ is viewed as the empirical $L_2$-norm. $a \simeq b$ means that there are two positive constants $c$, $C$ such that $ca \leq b \leq Ca$. Also, $a = O(b)$ means that there is some positive constant $C$ such that $a \leq Cb$ in probability, and $a = o(b)$ means that $a/b \to 0$ in probability. To ease the notations, we write $[M] := \{1, ..., M\}$.

## 2 Preliminaries and Algorithms

Given a compact set $\mathcal{X}$, we denote a *positive semidefinite kernel* on $\mathcal{X} \times \mathcal{X}$, i.e., a symmetric function $K : \mathcal{X} \times \mathcal{X} \to \mathbb{R}$ satisfying $\sum_{i,j=1}^n c_i c_j K(x_i, x_j) \geq 0$ for any $x_1, ..., x_n$ in $\mathcal{X}$ and $c_1, ..., c_n \in \mathbb{R}$. As showed in [2], a positive semidefinite kernel on $\mathcal{X}$ is associated with a unique Hilbert space $\mathcal{H}_K$ consisting of functions on $\mathcal{X}$, and $\mathcal{H}_K$ is called a RKHS associated with the kernel $K$. RKHS is known for its *reproducing property*, i.e., for any $f \in \mathcal{H}_K$ and $x \in \mathcal{X}$, $f(x) = \langle f, K_x \rangle_K$ with $K_x = K(x, \cdot)$. Another key property of RKHS is the *spectral theorem*, which assures that $K$ admits the following eigen-decomposition:

$$K(x, x') = \sum_{\ell \geq 1} \mu_\ell \psi_\ell(x) \psi_\ell(x'), \text{ for any } x, x' \in \mathcal{X}, \tag{2}$$

where $\mu_1 \geq \mu_2 \geq ... \geq 0$ are its eigenvalues and $\{\psi_\ell : \ell \geq 1\}$ are the corresponding eigenfunctions, leading to an orthogonal basis in $L_2(\rho_X)$. These two fundamental properties of RKHS will be the foundation of our theoretical analysis.

To characterize functional complexities, we first introduce several basic facts of the empirical processes defined in RKHS. On basis of (2), for any $\delta \in (0, 1]$, we define

$$\omega_n(\delta) := \left( \frac{1}{n} \sum_{\ell=1}^n \left( \mu_\ell \wedge \delta^2 \right) \right)^{1/2},$$

where $a \wedge b$ means $\min\{a, b\}$. For some constant $A > 0$, we define a quantity associated with $\omega_n(\delta)$ as

$$\epsilon(K) := \inf \left\{ \epsilon \geq \sqrt{\frac{A \log M}{n}} : \omega_n(\delta) \leq \epsilon\delta + \epsilon^2, \forall \delta \in (0, 1] \right\}.$$

The quantity $\epsilon(K)$ plays a key role in bounding the excess risk in RKHS [40, 20]. Given appropriate decay of $\mu_\ell$, the upper bound of $\epsilon(K)$ can be derived explicitly in Section 3.1.

Given the training sample $\{X_i, Y_i\}_{i=1}^n$ i.i.d. from $\mathcal{X} \times \mathcal{Y}$, the standard SVM with a single kernel [45] solves the following optimization task

$$\min_{f \in \mathcal{H}_K} \left\{ \frac{1}{n} \sum_{i=1}^n \phi(Y_i f(X_i)) + \lambda \|f\|_K^2 \right\}, \tag{3}$$

where $\lambda$ is a regularization parameter that balances the tradeoff between the empirical risk and the function complexity. The theoretical properties of the standard SVM has been extensively studied in literature; see the earlier work of [11, 39, 7].

In the context of MKL, the sparse multi-kernel SVM [34] equips the empirical hinge loss with the $\ell_1$ penalty,

$$\min_{f = \sum_{m=1}^M f_m, \, f_m \in \mathcal{H}_{K_m}} \left\{ \frac{1}{n} \sum_{i=1}^n \phi(Y_i f(X_i)) + \lambda \sum_{m=1}^M \|f_m\|_{K_m} \right\}. \tag{4}$$

This type of sparse MKL method has been studied extensively in [34, 3, 20, 16], and among others. Essentially, it can be regarded as an infinite-dimensional version of the Lasso-type penalization [43].

In this paper, we present a different approach from (4) to formulate the sparse multi-kernel SVM. For simplicity, we assume that $\sup_{x \in \mathcal{X}} |K_m(x, x)| \leq 1$ for all $m \in [M]$. We use a new $L_1$-type regularization term partially inspired by the additive mean models [32]. Denote the bounded ball of $\mathcal{H}^M$ by $\mathbb{B}^M := \left\{ f = \sum_{j=1}^M f_m : f_m \in \mathcal{H}_{K_m}, \|f_m\|_{K_m} \leq 1 \right\}$, the regularization term we adopt for the multiple-kernel SVM combines the empirical $L_2$-norms and RKHS-norms. Specifically, the proposed sparse multi-kernel SVM is formulated as

$$\hat{f} = \sum_{m=1}^M \hat{f}_m := \arg\min_{f \in \mathbb{B}_M} \left\{ \frac{1}{n} \sum_{i=1}^n \phi(Y_i f(X_i)) + \sum_{m=1}^M \lambda_m \sqrt{\|f_m\|_n^2 + \gamma_m \|f_m\|_{K_m}^2} \right\}, \tag{5}$$

where $\|f_m\|_n$ is used to enforce the sparsity in $\hat{f}$, whereas $\|f_m\|_{K_m}$ is used to enforce the smoothness of each $\hat{f}_m$. Here $(\lambda_m, \gamma_m)_{m=1}^M$ are the regularization parameters, which will be specified in our theoretical results. By finite representation of reproducing kernel, each additive estimator $f_m(\cdot) = \sum_{i=1}^n \alpha_i^m K_m(x_i, \cdot)$ for all $m = 1, ..., M$. A direct computation leads to

$$\sqrt{\|f_m\|_n^2 + \gamma_n \|f_m\|_K^2} = \sqrt{(\alpha^m)^T \tilde{K}_m \alpha^m}$$

where $\tilde{K}_m = \frac{K_m^2}{n} + \gamma_n K_m$. Here $K_m$ is the kernel matrix induced by $K_m$ at points $\{x_i\}_{i=1}^n$. Note that $\tilde{K}_m$ is a semi-definite matrix, which can be written as $\tilde{K}_m = A^2$ with some matrix $A$. Hence, our original learning scheme in Eq.(5) can be transformed into a group Lasso optimization [32], and there exists several efferent numerical algorithms for solving it, such as proximal methods and coordinate descent ones.

This penalty term significantly differs from other sparsity penalties for nonparametric models in literature, such as $\sum_{m=1}^M \|f_m\|_{K_m}$ [22] and $\sum_{m=1}^M (\lambda_1 \|f_m\|_n + \lambda_2 \|f_m\|_{K_m})$ [20, 37]. Although the former penalty often generates sparse solutions, it is difficult to establish its theoretical results, due to the fact that the $\|\cdot\|_K$-norm cannot fully reflect the marginal distribution information. The latter penalty has been proved to enjoy some theoretical properties in [37], which is equivalent to the mixed $L_1$-norm $\left( \sum_{m=1}^M \lambda_m \sqrt{\|f_m\|_n^2 + \gamma_m \|f_m\|_{K_m}^2} \right)$ in (5). Theoretically, based on empirical processes theory, our proposed approach can also achieve improved learning rates, like the penalized method with the penalty $\sum_{m=1}^M (\lambda_1 \|f_m\|_n + \lambda_2 \|f_m\|_{K_m})$, which has been considered in [20, 31, 37] and others.

**Remark**. In general, $\|\hat{f}\|_n = 0$ does not imply $f = 0$. However, in the case of any kernel-based minimization problem, $\|\hat{f}\|_n = 0$ always implies $f = 0$. Based on the reproducing property of Mercer kernel, $f(x) = \langle f, K_x \rangle_K$ for any $f \in H_K$, where $H_K$ is a reproducing kernel Hilbert space. In fact, if $\|\hat{f}\|_n = 0$ holds, that is, $\hat{f}(x_i) = 0$ for all $i = 1, \ldots, n$, by the reproducing property we have $\langle \hat{f}, K_{x_i} \rangle_K = 0$ for all $i$. Hence, $\hat{f}$ is orthogonal to the subspace $S_n := span\{K_{x_1}, ..., K_{x_n}\}$. On the other hand, using the reproducing property again, any solution $\hat{f}$ of kernel-based minimization problems has a finite representation within $S_n$. So we conclude $\hat{f} = 0$.

**Remark**. The use of the mixed norm regularization is mainly motivated by the following fact: i) the proposed estimation with the mixed norm can lead to very sharp learning rates; ii) the empirical norm $\|\cdot\|_n$ used for sparsity is much milder than $\|\cdot\|_K$.

There exists several related work on multi-kernel SVM. Under the setting of one-dimensional additive kernels, Christmann and Hable [12] constructed kernels for additive SVM and provided consistent and

statistically robust estimators under the fixed dimensional and non-sparse setting. Zhao and Liu [53] proposed a group Lasso penalty by means of finite-bases approximation to a RKHS, and particularly they developed an efficient accelerated proximal gradient descent algorithm and established oracle properties of the SVM under sparse ultra-high dimensional setting (e.g. $M = o(e^n)$). However, the additive SVM model may suffer from the lack of algorithmic flexibility and underfitting especially when the true model involves interaction effects. Similar concerns have been raised towards linear SVM as in [55, 21, 52].

Note that in the past decade, a lot of work on general MKL with logarithmic dependence on $M$ have emerged [18, 41, 20], yet their analysis requires the loss function to be strongly convex, which rules out the commonly-used hinge loss for SVM. We also note that [13] provided an upper bound $O_p(\sqrt{\log M/n})$ of the Rademacher complexity with the $L_1$-norm constraint, which may lead to the same decay rate of the excess risk of SVM. However, this rate is not tight in general, since it is known that the fast rate of order $1/n$ can be attained for the linear case [11, 44].

This paper primarily focuses on the non-asymptotic analysis of the proposed sparse multi-kernel SVM method in (5) with an exponential number of kernels. Under the best ideal settings, the relative classification error of the proposed method is of near-oracle rate $O(s \log(M)/n)$, as if we knew the true sparsity in advance. Moreover, the method is adaptive to the sparsity of the learning problem and the margin parameter. In our proof, we have to face some technical challenges, such as dealing with non-smoothness of the hinge loss, functional complexities and NP-dimensionality.

# 3 Main Results

This section quantifies the asymptotic behavior of the proposed sparse multi-kernel SVM (5) in estimating the oracle predictor $f^*$ in (1). Its asymptotic convergence rates are established in terms of both generalization error and estimation error.

**Assumption A** (Bernstein condition). There exist universal constants $c_0 > 0$ and $\kappa \geq 1$, such that

$$\mathcal{E}(f) - \mathcal{E}(f^*) \geq c_0 \|f - f^*\|_2^{2\kappa}, \quad \text{for all } f \in \mathbb{B}^M.$$

Assumption A is a lower bound for the hinge excess risk in term of the $L_2(\rho_X)$ norm, as a strong identification condition of the population quantity with the hinge loss. The Bernstein condition stems from [4] and it has been verified for the hinge loss in [1]. Particularly, for linear SVM, it has been verified in [36] that the Bernstein condition holds with $\kappa = 1$, leading to the fast learning rate. Related to the Bernstein condition, a more standard margin condition in [42] has been commonly assumed in literature, where $\mathcal{E}(f) - \mathcal{E}(f_c) \geq c_0 \|f - f_c\|_1^{2\kappa}$, and $f_c$ is the minimizer of the misclassification error over all possible measurable functions. As a consequence, the Bernstein condition is more stringent than the standard margin condition, and detailed discussion is referred to Proposition 8.3 of [1].

Since the complexity of a RKHS is determined by the decay rate of the eigenvalues $\mu_\ell$'s [39], we now introduce the following spectral condition for the subsequent analysis.

**Assumption B** (Spectral condition). There exist a sequence of $0 < \tau_m < 1$ and a universal constant $c_1 > 0$, such that

$$\mu_\ell^{(m)} \leq c_1 \ell^{-1/\tau_m}, \quad \forall \ell \geq 1, \, m \in [M]. \tag{6}$$

Eq.(6) means the decay rate of the eigenvalues of kernel is polynomial. Note that $\tau_m < 1$ is a very weak condition, due to the relation that $\sum_{\ell=1} \mu_\ell^{(m)} = \mathbb{E}[K_m(X, X)] \leq 1$. For example, if $\rho_X$ is the Lebesgue measure on $[0, 1]$, it is known that $\mu_\ell^{(m)} \asymp \ell^{-2\alpha}$ for the Sobolev class $\mathcal{H}_{K_m} = \mathcal{W}_2^\alpha$ with $\alpha > \frac{1}{2}$. Indeed, spectral condition has a close quantitative relationship with the entropy number of the RKHS under mild conditions; see [40] for details.

**Assumption C** (Sup-norm condition). For the sequence of $0 < \tau_m < 1$ given in Assumption B, there exists some universal constant $c_2 > 0$, such that

$$\|g\|_\infty \leq c_2 \|g\|_2^{1-\tau_m} \|g\|_{K_m}^{\tau_m}, \quad \forall g \in \mathcal{H}_{K_m}, \, m \in [M].$$

As pointed out in [40], under some mild conditions on $\mathcal{H}_{K_m}$'s, Assumption C is equivalent to the spectral decay, as stated in Assumption B.

For some constant $b > 0$, we define a restricted subset of $\mathcal{H}^M$ by

$$\mathcal{F}_S^b = \left\{ f \in \mathbb{B}_M : \sum_{m=1}^M \lambda_m \sqrt{\|f_m - f_m^*\|_2^2 + \epsilon^2(K_m)\|f_m - f_m^*\|_{K_j}^2} \right.$$

$$\left. \leq b \sum_{m \in S} \lambda_m \sqrt{\|f_m - f_m^*\|_2^2 + \epsilon^2(K_m)\|f_m - f_m^*\|_{K_j}^2} \right\}.$$

The set $\mathcal{F}_S^b$ is a cone in the space $\mathcal{H}^M$, where the components corresponding to $j \in S$ dominate the remaining ones.

The following quantity is also crucial in our theoretical analysis, which is used to describe how 'dependent' these different RKHS's are. Particularly,

$$\Gamma(S; \rho_X) := \sup \left\{ \gamma > 0 : \gamma \left( \sum_{m \in S} \|f_m - f_m^*\|_2^2 \right) \leq \left\| \sum_{m=1}^M (f_m - f_m^*) \right\|_2^2, (f_1, ..., f_M) \in \mathcal{F}_S^b \right\}.$$

We can regard $\Gamma(S; \rho_X)$ as a generalized correlation between the components corresponding to $j \in S$ and $j \in S^c$, respectively.

**Assumption D** (Correlation condition). There exists some universal constant $c_3$, such that

$$\Gamma(S; \rho_X) > c_3 > 0.$$

Loosely speaking, this represents the correlation among RKHS's over the cone set where the components within the relevant indices $S$ well "dominate" the remaining ones. Lemma 1 in [41] shows that Assumption D is related to two geometric quantities. In fact, Assumption D has been widely used for various sparse problems, such as [10] and [6] for linear models, and [20] and [41] for sparse MKL with the quadratic loss.

**Remark**. Assumptions A and D imply that $c_0^{1/\kappa} c_3 \sum_{m \in S} \|f_m - f_m^*\|_2^2 \leq \left( \mathcal{E}(f) - \mathcal{E}(f^*) \right)^{1/\kappa}$ over $\mathcal{F}_S^b$, which is sufficient for deriving most of our results except for the estimation error. We also observe that, for the high-dimensional linear SVM, the restricted eigenvalue condition in [36] implies the above conclusion.

## 3.1 Oracle Rates

When the oracle predictor $f^*$ defined over $\mathcal{H}^M$ is sparse, we now state the upper bounds on the excess risk and the estimation error of the proposed multi-kernel SVM in (5). We allow the number of kernels $M$ and the number of active kernels $s$ increases with the sample size $n$.

**Theorem 1.** *Suppose that Assumptions A, C and D hold, and all the following constraints are satisfied: $2M\epsilon(K_m) \leq e^M$, $\lambda_m \geq 4C_0 C_1 \epsilon(K_m)$ and $\gamma_m \geq 4\epsilon^2(K_m)/C_0^2$ for all $m \in [M]$. Then with probability at least $1 - 4M^{-A}$, the estimated $\ell_1$-norm SVM function $\hat{f}$ satisfies*

$$\mathcal{E}(\hat{f}) - \mathcal{E}(f^*) \leq \max \left\{ \left( \frac{4\sqrt{2c_0}C_0}{\Gamma(S, \rho_X)} \right)^{\frac{2\kappa}{2\kappa-1}} \left( \sum_{m \in S} \lambda_m^2 \right)^{\frac{\kappa}{2\kappa-1}}, 32 \sum_{m \in S} \lambda_m \sqrt{\gamma_m} \right\}.$$

*Additionally, there also holds*

$$c_0 \|\hat{f} - f^*\|_2^{2\kappa} \leq \max \left\{ \left( \frac{4\sqrt{2c_0}C_0}{\Gamma(S, \rho_X)} \right)^{\frac{2\kappa}{2\kappa-1}} \left( \sum_{m \in S} \lambda_m^2 \right)^{\frac{\kappa}{2\kappa-1}}, 32 \sum_{m \in S} \lambda_m \sqrt{\gamma_m} \right\},$$

*with the same probability as above. Here $A$, $C_0$ are positive constants specified in Lemma 1, and $C_1$ specified in Proposition 1 may depend on $A$ and $c_2$.*

The technical proof of Theorem 1 is given in Appendix A. It is easy to check that Theorems 1 also holds if one replaces $M$ in the by an arbitrary $\tilde{M} \geq M$ such that $\log \tilde{M} \geq 2 \log \log n$. In this case, the probability bounds in the theorems become $1 - 4\tilde{M}^{-A}$. It also has a number of corollaries,

obtained by specifying particular choices of kernels. As Assumption B does not require a lower bound of the spectral decay, so all the finite-dimensional RHKS's, Sobolev classes and Gaussian kernels are covered in our settings. We here only present a corollary for the RKHS's with infinite eigenvalues with decay rate as in Assumption B. In this case, the upper bound of $\epsilon(K_m)$ is given by

$$\epsilon(K_m) \simeq \left\{ \sqrt{\frac{\log M}{n}} \vee n^{-\frac{1}{2(1+\tau_m)}} \right\}.$$

In particular, this type of scaling covers Sobolev spaces, consisting of functions with $\lfloor \frac{1}{2\tau_m} \rfloor$ derivatives. Up to some constants, we now present a direct corollary from Theorem 1 in a homogeneous setting.

**Remark**. $(C_0, C_1)$ are two constants independent of $n, M$ or $s$. Their definitions rely on the result of Proposition 5 in [20], where their constant did not give an explicit form. So, we can not give a more explicit form on $(C_0, C_1)$. Since $\gamma_n = \lambda_n^2$ in our theory, there is no additional hyperparameter to be optimized. To explain the role of two hyperparameters, we rewrite the mixed penaltation with two different parameters as:

$$\lambda_n \sqrt{\|f_m\|_n^2 + \gamma_n \|f_m\|_K^2} = \sqrt{\beta_n \|f_m\|_n^2 + \theta_n \|f_m\|_K^2}.$$

We see from the above equation that, $\beta_n$ is used to control sparsity, while $\theta_n$ is used to control functional smoothness, due to the fact that $\theta_n$ is a smaller order of $\beta_n$, precisly, $\theta_n = \beta_n^2$.

**Corollary 1.** *Under the same conditions of Theorem 1 and Assumption B holds in that each kernel with eigenvalues decays at rate $\mu_\ell^{(m)} = O(\ell^{-1/\tau})$ for some common $\tau < 1$. Then any solution $\hat{f}$ to (5) with $\lambda_m \simeq \epsilon(K_m)$ and $\gamma_m \simeq \epsilon^2(K_m)$ for all $m \in [M]$ satisfies*

$$\max \left\{ \|\hat{f} - f^*\|_2^{2\kappa}, \mathcal{E}(\hat{f}) - \mathcal{E}(f^*) \right\} = \frac{s}{\Gamma^2(S, \rho_X)} O\left( n^{-\frac{1}{1+\tau}} + \frac{\log M}{n} \right)^{\frac{\kappa}{2\kappa-1}},$$

*with probability at least $1 - 4M^{-A}$. Specially for $\kappa = 1$, we have*

$$\max \left\{ \|\hat{f} - f^*\|_2^2, \mathcal{E}(\hat{f}) - \mathcal{E}(f^*) \right\} = \frac{s}{\Gamma^2(S, \rho_X)} O_p\left( n^{-\frac{1}{1+\tau}} + \frac{\log M}{n} \right).$$

Corollary 1 considers the homogeneous setting that all the RKHS's have the same complexities, denoted by the parameter $\tau$ in Assumption B. For the Gaussian kernel and the typical case with $\kappa = 1$, the parameter $\tau$ is close to zero and thus the excess risk of our estimator attains the order $O_p(s \log M/n)$ up to the term $\Gamma(S, \rho_X)$, which is the minimax rate of the least square parametric regression; see [37] for details.

It is worth noting that, the choices of the regularization parameters $(\lambda_m, \gamma_m)$ are adaptive to the sparsity and the margin, whereas the sparsity parameter $s$ and the Bernstein parameter $\kappa$ are not needed to learn the proposed estimator. Moreover, as stated in [20], $\tau_m$ can be replaced by its empirical estimator based on $\mathbf{K}_m = (K_m(x_l, x_k))_{l,k=1}^n$, this further implies that we can define two data-driven regularization parameters instead of $(\lambda_m, \gamma_m)$. Here we omit the details to avoid repetition.

**Remark**. In view of the popularity of SVM in machine learning, this paper focuses on theoretical investigation on the hinge loss with a mixed functional norm under multi-kernel setting. In fact, the current technical analysis can be easily extended to any Lipschitz loss case, e.g., the Huber loss and the quantile loss used for robust methods. Yet we think this is also beyond the focus of this paper.

## 3.2 Relative Classification Error

The goal of a binary classification procedure is to predict the label $Y \in \{-1, 1\}$ given the value of $X$. A binary classifier $f : \mathcal{X} \to \{-1, 1\}$ is a function from $\mathcal{X}$ to $\mathcal{Y}$ which divides the input space $\mathcal{X}$ into two classes. Let us split $\rho(X, Y) = \rho_X(X) \times \mathbb{P}(Y|X)$, where $\rho_X$ is the marginal distribution on $\mathcal{X}$ and $\mathbb{P}(\cdot|x)$ is the conditional probability measure given $X$. The efficiency of a binary classifier $f$ is measured by the so-called *misclassification error*

$$\mathcal{R}(f) := \mathbb{P}[f(X) \neq Y] = \int_{\mathcal{X}} \mathbb{P}[Y \neq f(x)|x] d\rho_X(x).$$

It is known that $f_c(x) = \text{sgn}(2\eta(x) - 1)$ is a minimizer of $\mathcal{R}(f)$ over all measurable functions, where $\eta(x) = \mathbb{P}(Y = 1 \,|\, X = x)$ is the conditional probability of $Y = 1$ given $x$. Thus, to assess the classification performance of a classifier $f$, its *relative classification error*, defined as $\mathcal{R}(f) - \mathcal{R}(f_c)$, is of some significance.

In empirical risk minimization, the optimization of misclassification error is difficult due to its non-convexity (i.e. $0/1$ loss), and a common strategy is to find a surrogate convex loss to replace the non-convex 0/1 loss, such as the hinge loss, the logistic loss, or the quadratic loss. Therefore, to quantify the classification error of a SVM classifier, it is natural to ask for the connection between the hinge loss and the $0/1$ loss. Recall from Theorem 9.21 in [14], for any measurable function $f : \mathcal{X} \to \mathbb{R}$, the following inequality holds

$$\mathcal{R}(\text{sgn}(f)) - \mathcal{R}(f_c) \leq \mathcal{E}(f) - \mathcal{E}(f_c). \tag{7}$$

Note that our general results are on the smooth function $f^*$ rather than on $f_c$, and then it is often impossible to provide rates on the estimation of $f_c$ without stringent assumption on $\mathbb{P}(Y|X)$ and $\mathcal{H}_s = \cup_{|S|=s}\mathcal{H}_S$ with $\mathcal{H}_S := \left\{ f = \sum_{m \in S} f_m, f_m \in \mathcal{H}_m, \|f_m\|_{K_m} \leq 1 \right\}$. Usually, $f_c$ is not necessarily sparse and smooth as $f^*$, we need to consider the approximation error between all the possible sparse multi-kernel spaces and $f_c$, defined by $\mathcal{A}(\mathcal{H}_s, f_c) := \inf_{f \in \mathcal{H}_s} \{\mathcal{E}(f) - \mathcal{E}(f_c)\}$. The quantity $\mathcal{A}(\mathcal{H}_s, f_c)$ measures the approximation error of $\mathcal{H}_s$ in approximating $f_c$. The sparsity $s$ balances the approximation error $\mathcal{A}(\mathcal{H}_s, f_c)$ and the effective dimension of the function class $\mathcal{H}_s$. Based on this notation, for any $f$ we can rewrite $\mathcal{E}(f) - \mathcal{E}(f_c) = \mathcal{E}(f) - \mathcal{E}(f^*) + \mathcal{E}(f^*) - \mathcal{E}(f_c) = \mathcal{E}(f) - \mathcal{E}(f^*) + \mathcal{A}(\mathcal{H}_s, f_c)$. This together with Theorem 1 and (7) leads to the upper bounds of the relative classification error.

**Theorem 2.** *Assume the same conditions of Theorem 1 are all met. We choose $\lambda_m \simeq \epsilon(K_m)$ and $\gamma_m \simeq \epsilon^2(K_m)$ for all $m \in [M]$, then with probability at least $1 - 4M^{-A}$, we have*

$$\mathcal{R}(\text{sgn}(\hat{f})) - \mathcal{R}(f_c) = O\left( \left( \frac{1}{\Gamma(S, \rho_X)} \right)^{\frac{2\kappa}{2\kappa-1}} \left( \sum_{m \in S} \epsilon^2(K_m) \right)^{\frac{\kappa}{2\kappa-1}} + \sum_{m \in S} \epsilon^2(K_m) \right) + \mathcal{A}(\mathcal{H}_s, f_c).$$

*In the homogeneous cases as Corollary 1, we also have*

$$\mathcal{R}(\text{sgn}(\hat{f})) - \mathcal{R}(f_c) = \frac{s}{\Gamma^2(S, \rho_X)} O\left( n^{-\frac{1}{1+\tau}} + \frac{\log M}{n} \right)^{\frac{\kappa}{2\kappa-1}} + \mathcal{A}(\mathcal{H}_s, f_c).$$

As mentioned earlier, ordinary kernels including the Gaussian kernel and the Laplace kernel are universal in $L^1_{\rho_X}$ in the sense that the approximation error $\mathcal{A}(\mathcal{H}_s, f_c)$ is negligible if $f_c \in L^1_{\rho_X}$, and in this case the excess risk of the estimator dominates the approximation error. Note that $\hat{f}^*$ is a sparse minimizer of $\mathcal{E}(\cdot)$ defined on the multi-kernel class $\mathcal{H}^M$. From a model selection point of view, we are mainly interested in the selection of different RKHS's, whereas the classical SVM in (3) focuses more on selection of parameters within a single RKHS; see [7] for details.

Similar to the common margin assumption in classification [11, 44, 1], the smaller Bernstein parameter $\kappa$ implies the lower noise level of $\eta(x)$ near $1/2$. Particularly, our fast rate in Theorem 2 equals $O\left( n^{-\frac{1}{1+\tau}} + \frac{\log M}{n} \right)$ when $\kappa = 1$. If there is no assumption on the margin ($\kappa \to \infty$), the rate is arbitrarily close to $O\left( n^{-\frac{1}{1+\tau}} + \frac{\log M}{n} \right)^{1/2}$ when $s$ is fixed, which matches the minimax lower bounds without any low noise condition on the margin [47].

### 3.3 Related Work

As mentioned in the introduction, the popularity of sparse multi-kernel classification is mainly due to its flexibility and increased interpretability, including single kernel-based SVM and sparse high-dimensional SVM as special cases. Hence, it is natural to compare the learning rate established in Theorem 1 and Corollary 1 over the multi-kernel class with some existing results in literature.

**I. Single kernel learning**

In [11], the regularization error and the projection operator are introduced to derive convergence rates of the misclassification error. In particular, Theorem 10 there states that for a separation exponent $\theta$,

the minimizer $\hat{f}$ of the standard regularized learning scheme (3) satisfies

$$\mathcal{R}(\text{sgn}(\hat{f})) - \mathcal{R}(f^*) = O_p\left(n^{-\frac{1}{1+2\tau/\theta+2\tau}}\right),\tag{8}$$

based on the relation between covering number and spectral decay. Our bound in Theorem 2 is always better than (8) even in their best case with $\theta \to +\infty$.

## II. Multiple kernel learning

In [46], they analyzed the general SVM scheme by varying scale parameter $\sigma$ of the Gaussian kernel, which can be regarded as a family of specific kernels. To be precise, the multi-kernel SVM in [46] is formulated by adding minimization on $\sigma$ to the original learning scheme (3). In particular, Theorem 6 there states that when no approximation error is involved, the minimizer $\hat{f}$ has the following error bound,

$$\mathcal{R}(\text{sgn}(\hat{f})) - \mathcal{R}(f^*) = O_p\left(n^{-\frac{\kappa}{2\kappa-1+\tau}}\right),\tag{9}$$

Note that when $\kappa \to 1$ corresponding to the lowest noise case, the error bound in (9) is of the best order $O_p(n^{-\frac{1}{1+\tau}})$, which is the same as that in our Theorem 2.

Besides, without specification of learning algorithms, Cortes et al. [13] provided refined generalization bounds for learning kernels based on a convex combination of $M$ basis kernels with $L_1$-regularization. When the hinge loss is used, Corollary 1 there can give the misclassification error with the order $(\log M/n)^{1/2}$, which is substantially slower than the error bound in our Theorem 2, derived by advanced empirical process techniques.

## III. High-dimensional learning

High dimensional data has attracted great interest in recent years, where the number of parameters (or dimensionality) can be much greater than the sample size. As mentioned earlier in Introduction, several work have considered high-dimensional linear or additive SVM classification.

(a) **Linear SVM.** Zhang et al. [52] proposed a sequence of non-convex penalized SVM's in moderately high dimension, and gave variable selection consistency and oracle property for a statistical perspective. Also, Peng et al. [36] investigated the statistical performance of the $L_1$-norm SVM in high dimension, and established refined error bound of its estimation error, with a near-oracle rate $O(\sqrt{s \log M/n})$, where $M$ is the number of candidate features and $s$ is the sparsity parameter. Note that the linear case corresponds to $\tau \to 0$ in the spectral assumption, and thus this rate in [36] is almost consistent with our derived result in Corollary 1.

(b) **Additive SVM.** Zhao and Liu [53] developed a sparse high dimensional nonparametric classification method with additive kernels, where each kernel $K_j$ is one-dimensional function defined on each coordinate. Also, in their Theorem 5.1, they provided the oracle properties of the estimation error for the sparse additive SVM,

$$\mathcal{E}(\hat{f}) - \mathcal{E}(f^*) = O_p\left(\tilde{\eta} + s\sqrt{\frac{q \log M}{n}}\right),\tag{10}$$

where $\tilde{\eta}$ is referred to as the approximation error and $q$ is the number of finite base approximation to the additive components. Even in the best case with $q = 1$, the convergence rate in (10) is not comparable to our oracle rate established in Theorem 1, not to mention that $q$ is often required to diverge at a polynomial order of $n$ for functional flexibility.

The relation between our analysis and existing analyses are given in the follow table.

| Method | Penalty | Optimization | Convergence rate |
|--------|---------|--------------|------------------|
| [13] | $\sum_{m=1}^{M} \|f_m\|_{K_m}$ | Group Lasso | $\log(M)/\sqrt{n}$ |
| [37] | $\sum_{m=1}^{M} \|f_m\|_n + \gamma_n \sum_{m=1}^{M} \|f_m\|_{K_m}$ | SOCP | $sn^{-\frac{1}{1+\tau}} + \frac{s \log M}{n}$ |
| [53] | $\sum_{m=1}^{M} \|f_m\|_2$ | Proximal GD | $s\sqrt{\frac{\log M}{n^{4/5}}}$ |
| This Paper | $\sum_{m=1}^{M} \sqrt{\|f_m\|_n^2 + \gamma_n\|f_m\|_{K_m}^2}$ | Group Lasso | $\left(sn^{-\frac{1}{1+\tau}} + \frac{s \log M}{n}\right)^{\frac{\kappa}{2\kappa-1}}$ |

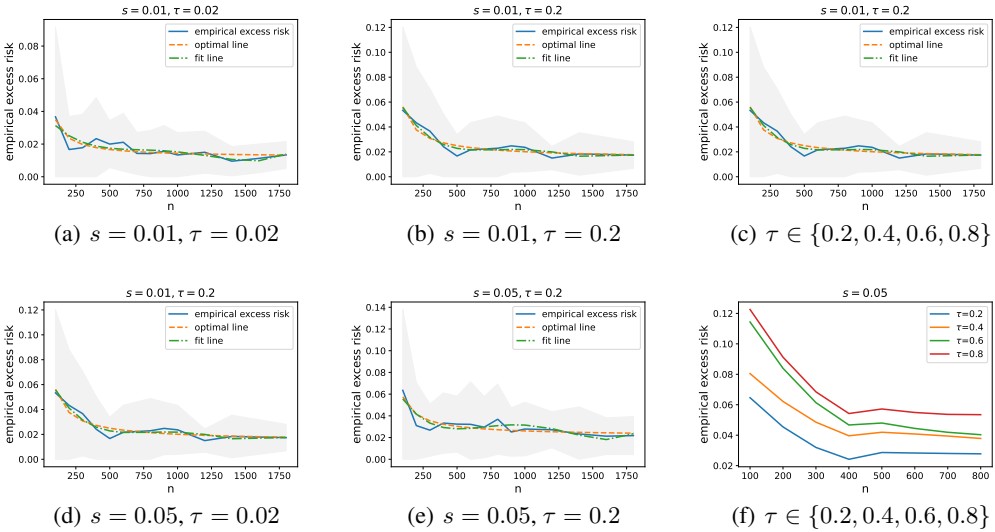

(a) $s = 0.01, \tau = 0.02$        (b) $s = 0.01, \tau = 0.2$        (c) $\tau \in \{0.2, 0.4, 0.6, 0.8\}$

(d) $s = 0.05, \tau = 0.02$        (e) $s = 0.05, \tau = 0.2$        (f) $\tau \in \{0.2, 0.4, 0.6, 0.8\}$

Figure 1: The empirical excess error of our method on the test set with different size of training data $n$, the sparse rate $s$ and the parameter $\tau$. In $(a, b, d, e)$, the blue line is the empirical excess error on the test set, the dotted orange line is the optimal rate of theoretical findings, the dotted green line is the fit curve of the empirical excess error.

## 4 Numerical Experiments

In this section, we report the results of our numerical experiments on simulated data aimed at validating our theoretical results. First we consider constructing multiple kernels by random features of the spline kernel. A spline kernel of order $q$ :

$$K_{2q}\left(\mathbf{x}, \mathbf{x}'\right) = 1 + \sum_{k=1}^{\infty} \cos\left(2\pi k\left(\mathbf{x} - \mathbf{x}'\right)\right) / \left(k^{2q}\right).$$

If the marginal distribution of $\mathcal{X}$ is uniform on $[0, 1]$, then $K_{2q}\left(\mathbf{x}, \mathbf{x}'\right) = \int_0^1 \psi(\mathbf{x}, \boldsymbol{\omega})\psi(\mathbf{x}, \boldsymbol{\omega})\varrho(\boldsymbol{\omega})d\boldsymbol{\omega}$, where $\psi(\mathbf{x}, \boldsymbol{\omega}) = K_q(\mathbf{x}, \boldsymbol{\omega})$ and $\varrho(\boldsymbol{\omega})$ is also uniform on $[0, 1]$. We sample uniformly $M$ times from $[0, 1]$, so we get $M$ kernels : $\psi\left(\mathbf{x}, \boldsymbol{\omega}_1\right), \dots, \psi\left(\mathbf{x}, \boldsymbol{\omega}_M\right)$. Then we construct the target function $f_*$ by randomly choose $Ms$ kernels and corresponding weights, where $0 < s < 1$ is the sparse rate. For a classification problem, we map the target value to $\{+1, -1\}$ labels.

We generate different size of samples, then split them into train set and test set. The number of kernels $M$ is set to 1000, the regularization parameters are set to be $\lambda = n^{-\frac{1}{2(1+\tau)}}, \gamma = n^{-\frac{1}{1+\tau}}$ as from the theoretical analysis of Corollary 1. We repeat the training 20 times and estimate the averaged hinge loss on test data. The averaged hinge loss on the test data with different size of train data is given in Figure 1. From Figure 1$(a, b)$ or $(d, e)$, we can see that the line of best fit for empirical excess risks match the learning rate $s \cdot n^{-\frac{1}{1+\tau}}$ (from Corollary 1), which verifies our theoretical findings. Figure 1$(c)$ or $(f)$ illustrates that at the same sparse rate $s$, the less the parameter $\tau$, the better the performance. One the other hand, Figure 1$(a, d)$ $(b, e)$ or $(c, f)$ shows that good performance can be obtained at a sparser setting. The above results is consistent with the theoretical analysis of Theorem 1 and Theorem 2.

## 5 Conclusion

We establish the first generalization error bounds for sparse multi-kernel SVMs, where the margin complexity term and the number of the potential kernels are considered carefully. The results we present significantly improve on previous results even for the additive or linear cases.

## Acknowledgment

This work is supported in part by the National Natural Science Foundation of China (No. 62076234, No.61703396, No.62106257), Beijing Outstanding Young Scientist Program NO.BJJWZYJH012019100020098, Intelligent Social Governance Platform, Major Innovation & Planning Interdisciplinary Platform for the "Double-First Class" initiative, Renmin University of China, China Unicom Innovation Ecological Cooperation Plan, Public Computing Cloud of Renmin University of China.

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
