# Supplementary Material for "Improved Learning Rates of a Functional Lasso-type SVM with Sparse Multi-Kernel Representation"

**Shaogao Lv[1], Junhui Wang[4], Jiankun Liu[5], Yong Liu[2,3]**[*]
[1]Nanjing Audit University
[2]Gaoling School of Artificial Intelligence, Renmin University of China, Beijing, China
[3]Beijing Key Laboratory of Big Data Management and Analysis Methods, Beijing, China
[4]City University of Hong Kong
[5]Institute of Information Engineering, Chinese Academy of Sciences
`lvsg716@nau.edu.cn, liuyonggsai@ruc.edu.cn`

## Appendix

**The organization of the proofs is as follows**. Frist, we show that the $\ell_1$-type penalty has as special feature, allowing one to avoid estimating explicitly the estimation error. Second, a new element in the proof is proved in empirical process theory, in which we use the strong convexity of the expected functional objective under the Bernstein condition to enter directly into local conditions. Third, we use the intrinsic correlation between different RKHSs to derive a key inequality, which is related to the left side one of our basic inequality. Based on the above three steps, we can establish a basic polynomial-type inequality, which immediately yields our general learning rates for SVM.

## Proofs

Given a function space $\mathcal{G}$, recall that the empirical Rademacher average on $\mathcal{G}$ is defined by

$$\widehat{\mathrm{R}}(\mathcal{G}) = \sup_{g \in \mathcal{G}} \left| \frac{1}{n} \sum_{i=1}^{n} \sigma_i g(z_i) \right|, \quad \forall \, g \in \mathcal{G},$$

where $\{\sigma_i\}$ are i.i.d. Rademacher variables. The population-level Rademacher complexity of $\mathcal{G}$ is given by $\mathrm{R}(\mathcal{G}) = \mathbb{E}_{\sigma,z}[\widehat{\mathrm{R}}(\mathcal{G})]$. The contractive property of $\mathrm{R}(\mathcal{G})$ is very useful in our theoretical analysis. That is, if $\varphi : \mathbb{R} \to \mathbb{R}$ is Lipschitz with constant $L_\varphi$ and satisfies $\varphi(0) = 0$, then

$$\mathrm{R}(\varphi \circ \mathcal{G}) \leq 2L_\varphi \mathrm{R}(\mathcal{G}).$$

In addition, we state the close relationship between $\| \cdot \|_n$ and $\| \cdot \|_2$ for functions in $\mathcal{H}_m$'s. The following Lemma 1 follows immediately from Theorem 4 and Proposition 5 in [2].

**Lemma 1.** *Suppose that $A \geq 1$ and $\log M \geq 2 \log \log n$. Then there exists a universal constant $C_0 > 0$ such that with probability at least $1 - M^{-A}$, for all $f_m \in \mathcal{H}_m$,*

$$\|f_m\|_2 \leq C_0 \big( \|f_m\|_n + \epsilon(K_m) \|f_m\|_{K_m} \big) \quad \text{and} \quad \|f_m\|_n \leq C_0 \big( \|f_m\|_2 + \epsilon(K_m) \|f\|_{K_m} \big).$$

**Lemma 2.** *(Concentration Theorem [1]) Let $Z_1, ..., Z_n$ be independent random variables with values in some space $\mathcal{Z}$ and let $H$ be a class of real-valued functions on $\mathcal{Z}$, satisfying for some positive constants $\eta_n$ and $\tau_n$,*

$$\|h\| \leq \eta_n, \quad \text{and} \quad \frac{1}{n} \sum_{i=1}^{n} var(h(Z_i)) \leq \tau_n^2, \quad \forall \, h \in H.$$

---

[*]Corresponding Author

35th Conference on Neural Information Processing Systems (NeurIPS 2021).

*Define* $\boldsymbol{Z} := \sup_{h \in H} \left| \frac{1}{n} \sum_{i=1}^{n} \big( h(Z_i) - \mathbb{E} h(Z_i) \big) \right|$. *Then for* $t > 0$

$$\mathbb{P}\Big( \boldsymbol{Z} \geq \mathbb{E}(\boldsymbol{Z}) + t\sqrt{2(\tau_n^2 + 2\eta_n \mathbb{E}(\boldsymbol{Z}))} + \frac{2\eta_n t^2}{3} \Big) \leq \exp[-nt^2].$$

For any given $\Delta_-, \Delta_+ > 0$, we define the subset of $\mathcal{H}^M$

$$\mathcal{F}_\Delta := \{ f \in \mathcal{H}^M : \sum_{m=1}^{M} \epsilon(K_m)\|f_m - f_m^*\|_2 \leq \Delta_-, \sum_{m=1}^{M} \epsilon^2(K_m)\|f_m - f_m^*\|_{K_m} \leq \Delta_+ \}.$$

The following Proposition 1 on concentration inequalities holds for general lipschitz-type losses, including the hinge loss. Let $\mathcal{E}_n(f) = \frac{1}{n} \sum_{i=1}^{n} \phi(Y_i f(X_i))$ be the empirical risk of $f$ with respect to the population-level quantity $\mathcal{E}(f)$.

**Proposition 1.** *Let $\mathcal{F}_\Delta$ be a measurable function subset defined as above. Suppose that condition C holds for each univariate $\mathcal{H}_m$. For some $A \geq 1$, with probability at least $1 - 2M^{-A}$, the following bound holds uniformly on $\Delta_- \leq e^M$ and $\Delta_+ \leq e^M$,*

$$\big| \mathcal{E}_n(f) - \mathcal{E}_n(f^*) - \big( \mathcal{E}(f) - \mathcal{E}(f^*) \big) \big| \leq C_1(\Delta_- + \Delta_+ + e^{-M}), \quad \forall f \in \mathcal{F}_\Delta.$$

The proof of Proposition 1 is similar to that in [3] for high dimensional quantile regression, and it is given in Appendix B for completeness.

**Proof of Theorem 1.**

Given a subset $S \subseteq [M]$, we write $I_S(f; \lambda, \gamma) = \sum_{m \in S} \lambda_m \sqrt{\|f_m\|_n^2 + \gamma_m \|f_m\|_{K_m}^2}$. Clearly $I_S(f; \lambda, \gamma)$ is a mixed norm of $f$ for any $\lambda_m > 0$ and $\gamma_m > 0$. By the definition of $\hat{f}$ in (**??**) and the sparsity assumption on $f^*$, we have

$$\mathcal{E}_n(\hat{f}) + I_M(\hat{f}; \lambda, \gamma) \leq \mathcal{E}_n(f^*) + I_S(f^*; \lambda, \gamma). \tag{1}$$

Recall that $I_M(f; \lambda, \gamma) = I_S(f; \lambda, \gamma) + I_{S^c}(f; \lambda, \gamma)$ for any $f$, and $I_{S^c}(f; \lambda, \gamma) = I_{S^c}(f - f^*; \lambda, \gamma)$ by the sparsity assumption on $f^*$. Then it follows from (1) and the triangle inequality that

$$\mathcal{E}_n(\hat{f}) + I_{S^c}(\hat{f} - f^*; \lambda, \gamma) \leq \mathcal{E}_n(f^*) + I_S(\hat{f} - f^*; \lambda, \gamma).$$

Adding $I_S(\hat{f} - f^*; \lambda, \gamma)$ to both sides of the last inequality, we obtain

$$\mathcal{E}_n(\hat{f}) + I_M(\hat{f} - f^*; \lambda, \gamma) \leq \mathcal{E}_n(f^*) + 2I_S(\hat{f} - f^*; \lambda, \gamma).$$

Simple algebra yields that

$$
\begin{aligned}
\mathcal{E}(\hat{f}) + I_M(\hat{f} - f^*; \lambda, \gamma) &\leq \mathcal{E}(f^*) + \big| \mathcal{E}_n(\hat{f}) - \mathcal{E}_n(f^*) - \big( \mathcal{E}(\hat{f}) - \mathcal{E}(f^*) \big) \big| \\
&\quad + 2I_S(\hat{f} - f^*; \lambda, \gamma).
\end{aligned}
\tag{2}
$$

We now bound the quantities $I_M(\hat{f} - f^*; \lambda, \gamma)$ and $I_S(\hat{f}_n - f^*; \lambda, \gamma)$ by their population-level terms, respectively. Note that with probability at least $1 - M^{-A}$, we have

$$
\begin{aligned}
I_M(\hat{f} - f^*; \lambda, \gamma) &\geq \frac{1}{\sqrt{2}} \sum_{m=1}^{M} \lambda_m \big( \|\hat{f}_m - f_j^*\|_2/C_0 + (\sqrt{\gamma_m} - \epsilon(K_m))\|\hat{f}_m - f_m^*\|_{K_m} \big) \\
&\geq \frac{1}{\sqrt{2}C_0} \sum_{m=1}^{M} \lambda_m \big( \|\hat{f}_m - f_m^*\|_2 + C_0/2\sqrt{\gamma_m}\|\hat{f}_m - f_m^*\|_{K_m} \big) \\
&\geq \frac{1}{\sqrt{2}C_0} \sum_{m=1}^{M} \lambda_m \sqrt{\|\hat{f}_m - f_m^*\|_2^2 + C_0^2/4\gamma_m\|\hat{f}_m - f_m^*\|_{K_m}^2}, \tag{3}
\end{aligned}
$$

where the first inequality follows from Lemma 1 and the subadditivity of $\sqrt{\cdot}$, and the second inequity is based on assumption $\sqrt{\gamma_m} \geq 2\epsilon(K_m)/C_0$ for any $m \in [M]$, and the third inequality follows from the subadditivity of $\sqrt{\cdot}$ as well. By the similar arguments as above, with probability at least $1 - M^{-A}$, we also have that

$$I_S(\hat{f} - f^*; \lambda, \gamma) \leq \sqrt{2}C_0 \sum_{m \in S} \lambda_m \sqrt{\|\hat{f}_m - f_m^*\|_2^2 + 2/C_0^2 \gamma_m\|\hat{f}_m - f_m^*\|_{K_m}^2}. \tag{4}$$

Therefore, substituting (3) and (4) into (2) yields that with probability at least $1 - 2M^{-A}$,

$$\mathcal{E}(\hat{f}) + \frac{1}{\sqrt{2}C_0} \sum_{m=1}^{M} \lambda_m \sqrt{\|\hat{f}_m - f_m^*\|_2^2 + C_0^2/4\gamma_m \|\hat{f}_m - f_m^*\|_{K_m}^2} \leq \mathcal{E}(f^*) + 2\sqrt{2}C_0 \times$$

$$\sum_{m \in S} \lambda_m \sqrt{\|\hat{f}_m - f_m^*\|_2^2 + 2/C_0^2 \gamma_m \|\hat{f}_m - f_m^*\|_{K_m}^2} + \left| \mathcal{E}_n(\hat{f}) - \mathcal{E}_n(f^*) - \left( \mathcal{E}(\hat{f}) - \mathcal{E}(f^*) \right) \right|. \quad (5)$$

It remains to bound the empirical process $\left| \mathcal{E}_n(\hat{f}) - \mathcal{E}_n(f^*) - \left( \mathcal{E}(\hat{f}) - \mathcal{E}(f^*) \right) \right|$, for which Proposition 1 is employed.

Since $\|f_m\|_\infty \leq \|f_m\|_{K_m} \leq 1$ for any $f = (f_1, ..., f_M) \in \mathbb{B}^M$, and $2M\epsilon(K_m) \leq e^M$ for any $m \in [M]$, the following bounds are satisfied

$$\sum_{m=1}^{M} \epsilon(K_m) \|\hat{f}_m - f_m^*\|_2 \leq e^M, \quad \sum_{m=1}^{M} \epsilon^2(K_m) \|\hat{f}_m - f_m^*\|_{K_m} \leq e^M,$$

and thus Proposition 1 can be applied directly. Precisely, we obtain from Proposition 1 and (5) that, with probability at least $1 - 4M^{-A}$,

$$\mathcal{E}(\hat{f}) + \frac{1}{\sqrt{2}C_0} \sum_{m=1}^{M} \lambda_m \sqrt{\|\hat{f}_m - f_m^*\|_2^2 + C_0^2/4\gamma_m \|\hat{f}_m - f_m^*\|_{K_m}^2}$$

$$\leq \mathcal{E}(f^*) + 2\sqrt{2}C_0 \sum_{m \in S} \lambda_m \sqrt{\|\hat{f}_m - f_m^*\|_2^2 + 2/C_0^2 \gamma_m \|\hat{f}_m - f_m^*\|_{K_m}^2}$$

$$+ \sqrt{2}C_1 \sum_{m=1}^{M} \epsilon(K_m) \sqrt{\|\hat{f}_m - f_m^*\|_2^2 + \epsilon^2(K_m) \|\hat{f}_m - f_m^*\|_{K_m}^2} + C_1 e^{-M}. \quad (6)$$

With the choice of $\lambda_m \geq 4C_0 C_1 \epsilon(K_m)$ and $\gamma_m \geq 4\epsilon^2(K_m)/C_0^2$ for any $m \in [M]$, the above inequality immediately implies that

$$\mathcal{E}(\hat{f}) - \mathcal{E}(f^*) + \frac{1}{2\sqrt{2}C_0} \sum_{m=1}^{M} \lambda_m \sqrt{\|\hat{f}_m - f_m^*\|_2^2 + C_0^2/4\gamma_m \|\hat{f}_m - f_m^*\|_{K_m}^2}$$

$$\leq 2\sqrt{2}C_0 \sum_{m \in S} \lambda_m \sqrt{\|\hat{f}_m - f_m^*\|_2^2 + 2/C_0^2 \gamma_m \|\hat{f}_m - f_m^*\|_{K_m}^2} + C_1 e^{-M}. \quad (7)$$

We first consider the case when

$$(i): \ 2\sqrt{2}C_0 \sum_{m \in S} \lambda_m \sqrt{\|\hat{f}_m - f_m^*\|_2^2 + 2/C_0^2 \gamma_m \|\hat{f}_m - f_m^*\|_{K_m}^2} \geq C_1 e^{-M}.$$

It follows from (7) that

$$\mathcal{E}(\hat{f}) - \mathcal{E}(f^*) + \frac{1}{2\sqrt{2}C_0} \sum_{m=1}^{M} \lambda_m \sqrt{\|\hat{f}_m - f_m^*\|_2^2 + C_0^2/4\gamma_m \|\hat{f}_m - f_m^*\|_{K_j}^2}$$

$$\leq 4\sqrt{2}C_0 \sum_{m \in S} \lambda_m \sqrt{\|\hat{f}_m - f_m^*\|_2^2 + 2/C_0^2 \gamma_m \|\hat{f}_m - f_m^*\|_{K_m}^2}. \quad (8)$$

Since $\mathcal{E}(\hat{f}) - \mathcal{E}(f^*) \geq 0$ by definition, (8) implies that

$$\sum_{m=1}^{M} \lambda_m \sqrt{\|\hat{f}_m - f_m^*\|_2^2 + C_0^2/4\gamma_m \|\hat{f}_m - f_m^*\|_{K_m}^2}$$

$$\leq 16C_0^2 \sum_{m \in S} \lambda_m \sqrt{\|\hat{f}_m - f_m^*\|_2^2 + 2/C_0^2 \gamma_m \|\hat{f}_m - f_m^*\|_{K_m}^2}, \quad (9)$$

with probability at least $1 - 4M^{-A}$, provided that $\lambda_m \geq 4C_0 C_1 \epsilon(K_m)$ and $\gamma_m \geq 4\epsilon^2(K_m)/C_0^2$ for any $m \in [M]$. That is, $\hat{f}_n$ belongs to $\mathcal{F}_S^{16C_0^2}$ with high probability under the case (i). Meanwhile, from (8) we also conclude that, with probability at least $1 - 4M^{-A}$,

$$
\begin{aligned}
\mathcal{E}(\hat{f}_n) - \mathcal{E}(f^*) &\leq 4\sqrt{2}C_0 \sum_{m \in S} \lambda_m \|\hat{f}_m - f_m^*\|_2 + 16 \sum_{m \in S} \lambda_m \sqrt{\gamma_m} \\
&\leq 4\sqrt{2}C_0 \Big(\sum_{m \in S} \lambda_m^2\Big)^{1/2} \Big(\sum_{m \in S} \|\hat{f}_m - f_m^*\|_2^2\Big)^{1/2} + 16 \sum_{m \in S} \lambda_m \sqrt{\gamma_m}, \quad (10)
\end{aligned}
$$

for any $\hat{f} \in \mathcal{F}_S^{16C_0^2}$, where the second inequality follows from the Cauchy-Schwartz inequality. Under the identifiable assumption (Condition A) and the correlation assumption (Condition D), it follows from (10) that

$$
\begin{aligned}
\mathcal{E}(\hat{f}) - \mathcal{E}(f^*) &\leq 4\sqrt{2}C_0/\Gamma(S, \rho_X) \Big(\sum_{m \in S} \lambda_m^2\Big)^{1/2} \Big\| \sum_{m=1}^M (\hat{f}_m - f_m^*) \Big\|_2 + 16 \sum_{m \in S} \lambda_m \sqrt{\gamma_m} \\
&\leq 4\sqrt{2c_0}C_0/\Gamma(S, \rho_X) \Big(\sum_{m \in S} \lambda_m^2\Big)^{1/2} \big(\mathcal{E}(\hat{f}_n) - \mathcal{E}(f^*)\big)^{1/(2\kappa)} + 16 \sum_{m \in S} \lambda_m \sqrt{\gamma_m}, \quad (11)
\end{aligned}
$$

where the first inequality follows from Condition D, and the second inequity follows immediately from Condition A. Direct calculation of (11) yields that

$$
\mathcal{E}(\hat{f}) - \mathcal{E}(f^*) \leq \max \Big\{ \Big(\frac{4\sqrt{2c_0}C_0}{\Gamma(S, \rho_X)}\Big)^{\frac{2\kappa}{2\kappa-1}} \Big(\sum_{m \in S} \lambda_m^2\Big)^{\frac{\kappa}{2\kappa-1}}, 32 \sum_{m \in S} \lambda_m \sqrt{\gamma_m} \Big\}. \quad (12)
$$

Thus we complete the proof of Theorem 1 under case (i).

It remains to consider the case when (i) does not hold. That is,

$$
2\sqrt{2}C_0 \sum_{m \in S} \lambda_m \sqrt{\|\hat{f}_m - f_m^*\|_2^2 + 2/C_0^2 \gamma_m \|\hat{f}_m - f_m^*\|_{K_m}^2} < C_1 e^{-M}.
$$

It immediately follows from (7) that

$$
\mathcal{E}(\hat{f}) - \mathcal{E}(f^*) + \frac{1}{2\sqrt{2}C_0} \sum_{m=1}^M \lambda_m \sqrt{\|\hat{f}_m - f_m^*\|_2^2 + C_0^2/4 \gamma_m \|\hat{f}_m - f_m^*\|_{K_m}^2} \leq 2C_1 e^{-M}. \quad (13)
$$

It is clear that our desired result still holds, since $\log M \geq 2 \log \log n$ by assumption. Therefore, by combining (12) with (13), we complete the proof of Theorem 1.

**Appendix B: Proof of Proposition 1.**

To apply Theorem 2, denote $H = \{h(z) \mid h(z) = \phi(yf(x)) - \phi(yf^*(x)), \; f \in \mathcal{F}_\Delta\}$, where $\phi(u)$ is the hinge loss defined as above. We can write $[\mathcal{E}(f) - \mathcal{E}(f^*)] - [\mathcal{E}_n(f) - \mathcal{E}_n(f^*)] = \mathbb{E}[h(z)] - \frac{1}{n} \sum_{i=1}^n h(z_i), h \in H$. Then, by Bousquet's concentration inequality, with probability at least $1 - e^{-t}$,

$$
\boldsymbol{Z} \leq \mathbb{E}(\boldsymbol{Z}) + \sqrt{\frac{2t(\tau_n^2 + 2\eta_n \mathbb{E}\boldsymbol{Z})}{n}} + \frac{2\eta_n t}{3n}. \quad (14)
$$

The remaining proof is to give tight upper bounds of $\eta_n$, $\tau_n$ and $\mathbb{E}(\boldsymbol{Z})$ respectively. First, the sub-additivity of $\sqrt{\cdot}$ implies that

$$
\sqrt{\frac{2t(\tau_n^2 + 2\eta_n \mathbb{E}\boldsymbol{Z})}{n}} \leq \sqrt{\frac{2t}{n}} \tau_n^2 + 2\sqrt{\frac{\eta_n}{n} \mathbb{E}(\boldsymbol{Z})} \leq \sqrt{\frac{2t}{n}} \tau_n^2 + \mathbb{E}\boldsymbol{Z} + \frac{\eta_n}{n},
$$

where we used the basic inequality $\sqrt{uv} \leq (u + v)/2$ for any $u, v \geq 0$. Meanwhile, since $|Y| \leq 1$, the contraction property of $\phi$ implies $\mathbb{E}(h^2(Z)) \leq \|f - f^*\|_2^2$ for any $f \in \mathcal{F}_\Delta$. That is, $\tau_n^2 \leq \sup_{f \in \mathcal{F}_\Delta} \|f - f^*\|_2^2$. This together with (14) leads to

$$
\boldsymbol{Z} \leq 2\mathbb{E}(\boldsymbol{Z}) + \sqrt{\frac{2t}{n}} \sup_{f \in \mathcal{F}_\Delta} \|f - f^*\|_2 + \frac{(1+t)\eta_n}{n}. \quad (15)
$$

Moreover, by the contraction property of $\phi$ and Condition C, we have

$$\|h\|_\infty \leq \|f - f^*\|_\infty \leq \sum_{m=1}^M \|f_m - f_m^*\|_\infty \leq c_2 \sum_{m=1}^M \left(\|f_m - f_m^*\|_2 + \|f_m - f_m^*\|_{K_m}\right), \forall f \in \mathcal{F}_\Delta,$$

where we used the Young inequality $u^\tau v^{1-\tau} \leq u + v$ for any $u, v \geq 0$ and $0 < \tau \leq 1$. Note that $\epsilon(K_m) \geq \sqrt{\frac{A \log M}{n}}$ for all $m$, this follows that

$$\sum_{m=1}^M \|f_m - f_m^*\|_2 \leq \sqrt{\frac{n}{A \log M}} \Delta_-, \tag{16}$$

for any $f \in \mathcal{F}_\Delta$. A similar argument leads to $\sum_{m=1}^M \|f_m - f_m^*\|_{K_m} \leq \frac{n}{A \log M} \Delta_+$. So we combine these derived inequalities to obtain

$$\eta_n \leq \sqrt{\frac{c_2^2 n}{A \log M}} \Delta_- + \frac{c_2 n}{A \log M} \Delta_+. \tag{17}$$

Thus, plugging the upper bounds of $\eta_n$ (17) and $\sup_{f \in \mathcal{F}_\Delta} \|f - f^*\|_2$ (16) into (15), with probability at least $1 - e^{-t}$, we have

$$\boldsymbol{Z} \leq 2\mathbb{E}(\boldsymbol{Z}) + \sqrt{\frac{2c_2^2 t}{A \log M}} \Delta_- + \frac{c_2 n}{A \log M} \frac{(1+t)}{n} \Delta_+. \tag{18}$$

To bound $\mathbb{E}(\boldsymbol{Z})$, we use a symmetrization technique, and thus $\mathbb{E}(\boldsymbol{Z}) \leq 2\mathbb{E}[\widehat{R}(H)] \leq 2\mathbb{E}[\widehat{R}(\mathcal{F}_\Delta - f^*)]$, where the second inequality follows from the contraction property of Redemacher process. Moreover, applying Talagrand's concentration inequality [1] again for $\widehat{R}(\mathcal{F}_\Delta - f^*)$, we get that

$$\mathbb{E}[\widehat{R}(\mathcal{F}_\Delta - f^*)] \leq 2\widehat{R}(\mathcal{F}_\Delta - f^*) + \sqrt{\frac{2c_2^2 t}{A \log M}} \Delta_- + \frac{c_2 n}{A \log M} \frac{(1+t)}{n} \Delta_+,$$

with probability at least $1 - e^{-t}$. According to the existing result on weight empirical process in [2] (see Equation (8) below), on some event $E$ of probability at least $1 - M^{-A}$, for all $m \in [M]$ we have

$$\frac{1}{n} \left| \sum_{i=1}^n \sigma_i(f_m - f_m^*)(x_i) \right| \leq \tilde{C}\left[\epsilon(K_m)\|f_m - f_m^*\|_2 + \epsilon^2(K_m)\|f_m - f_m^*\|_{K_m}\right]. \tag{19}$$

Hence, with probability at least $1 - 2e^{-t} - M^{-A}$, we have

$$\boldsymbol{Z} \leq 8\widehat{R}(\mathcal{F}_\Delta - f^*) + 9\sqrt{\frac{2c_2^2 t}{A \log M}} \Delta_- + \frac{9c_2 n}{A \log M} \frac{(1+t)}{n} \Delta_+$$

$$\leq 8\sum_{j=1}^M \widehat{R}(\mathcal{H}_j - f_j^*) + 9\sqrt{\frac{2c_2^2 t}{A \log M}} \Delta_- + \frac{9c_2 n}{A \log M} \frac{(1+t)}{n} \Delta_+$$

$$\leq 8\tilde{C}\left(\Delta_- + \Delta_+\right) + 9\sqrt{\frac{2c_2^2 t}{A \log M}} \Delta_- + \frac{9c_2 n}{A \log M} \frac{(1+t)}{n} \Delta_+,$$

which holds on the event $E \cap F(\Delta_-, \Delta_+, t)$, where $\mathbb{P}(F(\Delta_-, \Delta_+, t)) \geq 1 - 2e^{-t}$. With the choice of $t = A \log M / c_2^2 + 4 \log M$, we obtain a bound that uniformly over

$$e^{-M} \leq \Delta_- \leq e^M \quad \text{and} \quad e^{-M} \leq \Delta_+ \leq e^M. \tag{20}$$

For this purpose, we consider $M^2$-different discrete pairs $\Delta_-^m = \Delta_+^m := 2^{-m}, \quad m \in [M]$. Then on the event $\bigcap_{k,m} F(\Delta_-^m, \Delta_+^k, t)$, we have $\boldsymbol{Z} \leq c(\Delta_-^m + \Delta_+^k)$ for all $m, k \in [M]$. Moreover,

$$\mathbb{P}\left( \bigcap_{k,m} F(\Delta_-^m, \Delta_+^k, t) \right) \geq 1 - 2M^2 e^{-c_2 \log M - 4 \log M} \geq 1 - 2M^{-2-c_2},$$

which tends to 1 as $M$ goes to infinity. Besides, using monotonicity of the functions $\Delta_-^m$ and $\Delta_+^k$ involved in the inequalities, the result can be extended to the whole range of $\Delta_-$ and $\Delta_+$ satisfying (20).

If $\Delta_- \leq e^{-M}$ or $\Delta_+ \leq e^{-M}$, it is trivial to derive the desired result with the same probability. This completes the proof of Proposition 1. $\qquad\square$