# OpenReview forum: "Improved Learning Rates of a Functional Lasso-type SVM with Sparse Multi-Kernel Representation"
_NeurIPS.cc/2021/Conference — NeurIPS 2021 Spotlight_

### Official Review · Reviewer_9Qmw · 2021-07-10

**Rating:** 8
**Confidence:** 5

**Summary:**

This paper focus on the theoretical analysis of SVM with sparse multi-kernel approximation, where the total number of kernels may be larger than the sample size. More specifically, the refined bounds for the excess risk on multi-kernel SVM are provided, including high-dimensional linear or additive classification as special cases. The oracle rates of convergence of classifiers depend on the complexity of multi-kernels, the sparsity, the Bernstein condition and the sample size, which significantly improves the previous results even for the additive or linear cases.

**Ethical Concerns:**

None.

**Limitations And Societal Impact:**

Here are some concerns:
1. It would be better to briefly give the sketch of the proof in the main body, especially some important lemma. For example, in Theorem 1 (Line 204), the paper claims "A, C0 are positive constants specified in Lemma 1". However, Lemma 1 only appears in Appendix and is not provided in the main body.
2.  It would be better to give a table to clearly depict the advantage compared with the related results: such as single kernel learning, multiple kernel learning and high-dimensional learning.
3. Some notations should be explained after their first appearance.  For example, in the equation after Line 108, what does the operator $\wedge$ means? In Eq.(6), what does $\mu_{l}^{(m)}$ means?
4. The font size in Figure 1 is too small to be readable.

**Main Review:**

In this paper,  a refined excess risk bound depends on the complexity of multi-kernels, the sparsity, the Bernstein condition and the sample size is provided, which significantly improves the previous results even for the additive or linear cases.  The oracle rate of the relative misclassification error is established, which consists of the same rate as the excess risk of the estimator plus some additional sparse approximation error. Many existing results can be incorporated as special cases of MKL for SVMs, so several existing results can be improved by the derived results. Moreover, the paper not only provides unified theoretical results for multi-kernel SVMs, but also enriches the literature on high-dimensional nonparametric classification. At last, the experiments on simulated data for SVM with sparse multi-kernel validate the theoretical findings.

In my opinion,  these results seem very interesting and important: the paper not only provides unified theoretical results for multi-kernel SVMs but also enriches the literature on high-dimensional nonparametric classification. Overall, I think this is a good theoretical and well-written paper, which is clearly of interest for the machine learning community.



**Time Spent Reviewing:**

3 days.

---

> ### Author Response · Authors · 2021-08-09
> **Response to Reviewer 9Qmw**
>
> Thanks for your valuable comments and suggestions.
>
> 1. Thanks for your valuable suggestions.  The organization of the proofs is as follows. Frist, we  show that  the $\ell_1$-type penalty has as special feature,  allowing one to avoid estimating explicitly the estimation error. Second, a new element in the proof is proved in empirical process theory, in which we use the strong convexity of the expected functional objective under the Bernstein condition to enter directly into local conditions. Third, we use the intrinsic correlation between different RKHSs to derive a key inequality, which is related to the left side one of our basic inequality.  Based on the above three steps, we can establish a basic polynomial-type inequality, which immediately yields our general learning rates for SVM.
> We will add this proof sketches in the final version.
>
> 2.  Thanks for your valuable suggestions. Considering that there exist too many related literatures on multiple kernel and high dimensional learning, we only list the most related work restricted to the sparse multi kernel learning framework. The relation between our analysis and existing analyses are given in the follow table.
>
> | Method  | Penalty| Optimization| Convergence rate|
> |:-------:|:--------:|:--------:|:--------:|
> |Cortes et.al(2010)|   $\sum_{m=1}^M\|f_m\|_{K_m}$                                 | Group Lasso  | $\log (M)/\sqrt{n}$ |
> |Raskutti et.al(2012) | $\sum_{m=1}^M\|f_m\|\_{n}+\gamma_n\sum_{m=1}^M\|f_m\|_{K_m}$ | SOCP         | $sn^{-\frac{1}{1+\tau}}+\frac{s\log M}{n} $|
> |Zhao and Liu(2012)   | $\sum_{m=1}^M\|f_m\|_{2}$                                   | Proximal GD  | $s\sqrt{\frac{\log M}{n^{4/5}}}$          |
> |This Paper|           $\sum_{m=1}^M\sqrt{\|f_m\|\_{n}^2+\gamma_n\|f_m\|^2_{K_m}}$  | Group Lasso  | $\Big(sn^{-\frac{1}{1+\tau}}+\frac{s\log M}{n}\Big)^{\frac{\kappa}{2\kappa-1}} $ |
>
> 3.  Thanks for your valuable suggestions.  $ a \wedge b$ means $\min\{a,b\}$. Eq.(6) means the decay rate of the eigenvalues of kernel is polynomial. We will explain the notations after their first appearance.
>
> 4. We will increase the font size of Figure 1.

---

### Official Review · Reviewer_MGcL · 2021-07-17

**Rating:** 7
**Confidence:** 3

**Summary:**

The paper proposes a new sparse multi-kernel SVM which utilizes a mixed L-2 norm to enforce the sparsity and a RKHS-norm to enforce smoothness, and theoretically analyzes the estimation bound and excess risk. Finally, the numerical experiments are also performed in this paper.

**Limitations And Societal Impact:**

No efficient optimization method will hinder the usage in practice.

**Main Review:**

Although the paper show that the proposed new formulation of the sparse MKL SVM enjoys better learning rate, it seems difficult to optimize due to the $\sqrt{}$ operator in regularization. Thus, to achieve the same generalization performance, few examples are needed, but maybe more computation time will be spent on optimization. On the whole, the new formulation does not bring too many benefits.

The experiments are performed on synthetic data, it seems better to show more results on real world data.

======================

Since the response has addressed my main concerns, I raise my score to 7.


**Time Spent Reviewing:**

48

---

> ### Author Response · Authors · 2021-08-09
> **Response to Reviewer MGcL**
>
> 1. Multiple kernel learning (MKL) with the single regularization $\|\cdot\|_K$ can be transformed into a group Lasso type optimization,  and there exist some efficient numerical algorithms to search approximate solutions, e.g. coordinate descent ones. However, these conventional MKL with $\|\cdot\|_K$ can not yeild sharp learning rates in general, mainly because an additional empirical norm $\|\cdot\|_n$ appears in a  weight empirical process, which is very crucial to faster learning rates under multi-kernel setting. To this end, we propose a new formulation of MKL with a mixed norm $\sqrt{\|f_m\|_n^2+\gamma_n\|f_m\|^2_K}$, and we can obtain sharp learning rates via the proposed method for multi-kernel SVM. Moreover, by the decomposition of a semi-definite matrix, the mix norm can also be transformed into a group Lasso-type one, analogues to the $\|\cdot\|_K$-based MKL. See more detailed discussions in F. Bach, JMLR, (2008) and Meier,  de Geer, and  Bühlmann, AOS, (2009). So, those existing algorithms for group Lasso can be used directly in our proposed SVM.
>
> 2. Some experiments  on real world datasets are added. We use two publicly available datasets, mushrooms and splice, from the LIBSVM Data.
> The accuracy and run time  of L1 sparse MKL and our mix-norm sparse MKL are given in the following table. We can find that our mix-norm sparse MKL outperforms L1 sparse MKL on accuracy with close time cost.
>
> | Method| Accuracy /Time | mushrooms| splice|
> |:-------:|:--------:|:--------:|:--------:|
> |L1 Sparse MKL| Accuracy |0.973| 0.887|
> |L1 Sparse MKL| Time |  11.212(s) |  1.547(s) |
> |Mix-norm Sparse MKL |Accuracy | 0.985|0.896|.
> |Mix-norm Sparse MKL| Time | 11.824(s)  |  1.571(s) |

---

### Official Review · Reviewer_xwyd · 2021-07-18

**Rating:** 8
**Confidence:** 2

**Summary:**

The paper consists of a theoretical analysis for bounds on the excess risk and estimation of SVMs in a sparse MKL setting.
Although the analysis is established for a larger class of combination of kernels with the Hinge loss, the derived bounds improve over existing ones, in terms of rates and dependency on the number of kernels, even in the more restrictive additive and linear cases and despite being adaptive to the unknown true sparsity.
It is worth noting that the analysis relies on the miniminization of the SVM problem with a specific penalty term that is slightly different from the ones used in previous work.
After deriving those theoretical results and comparing them to related work, the authors also provide some numerical experiments on simulated data whose generative process ensure that the true sparsity of the kernel combination is controlled.
The experimental results seem to closely verify the theoretical findings.

**Limitations And Societal Impact:**

Despite some comparison to related work, the limitations of this work have not been very deeply addressed. Addressing some of the remarkes I wrote in my main review would maybe improve that a bit.
There was no mention of potential negative societal impact but I am not sure that contribution in statistical learning theory have so much.

**Main Review:**

First of all, I want to acknowledge that I am not an expert reviewer for this submission.
Despite having contributed on MKL and in learning theory a long time ago, I have been mostly unaware of any development in the last decade or so.
Specifically, despite some efforts, I am not very confident about my ability to check the corectness of the proofs (although I could not spot any mistake).
Perhaps more importantly, I find myself to be struggling to carefully interpret or comment on the assumptions that are made to derive the analysis, despite efforts from the authors to relate them to assumptions that are made in related work.

I find the paper to be remarkably well-written (despite a few typos and mistakes I'll list later), and relatively easy to follow despite the technicality of the contributions.
Especially, the efforts made to write the proofs should be noted, especially as the proofs are "only" provided in the appendix.
I only have a few remarks or questions:
1) It is not clear to me how much the improvements over the existing results are specifically due to the choice of this peculiar penalty term. For instance, it would be nice to see whether the use of penalty terms that have been more classically used in the literature lead to slower convergence in the experimental setting used here.
2) Although it is mentioned that the chosen penalty term can "easily lead to a single adaptive group Lasso by the Cholesky decomposition", there is not any other comment about the algorithm, or computational complexity of solving problem (5). Can the authors comment? Or does it compare to formulations more commonly used?
3) Related to those last points, even if using that penalty term really improves the excess risk convergence in practive (beyond making this tighter analysis possible), can the authors comment on whether there is an interest for practioners to adopt (5)? More specifically, do you have any insight whether the (perhaps) faster empirical rate "compensate" for the higher computational cost induced by solving (5) rather than more computationally amenable formulations?

Here are some typos / mistakes I noticed in the paper:
- l.6: "and" was used twice
- l.174 "there existS"
- l.246 "multi-kernel cLAss", "From *a* model selection POV"
- l.254 "in this paper" was used twice
- l.259 "the popularity of sparse MK classification *is* mainly due"
- l.308 "from randomly choosING"
- l.323 "significantly improvE"

Last, I think recalling the theorems / propositions before exposing their proofs in the appendix would improve even further the ease of the reader.

**Time Spent Reviewing:**

8

---

> ### Author Response · Authors · 2021-08-09
> **Response to Reviewer xwyd**
>
> Thanks for your valuable comments and suggestions.
>
> 1. Thanks for your detailed comments. Theoretically, those multiple kernel learning with the commonly-used kernel norm penalty $\sum_{m=1}^M\|f_m\|_{K_m}$ often lead to a slow convergence rate, see M. Kloft and G. Blanchard., JMLR, (2012) for details. By contrast, using the mix norm  $\sqrt{\|f_m\|_n^2+\gamma_n\|f_m\|^2\_{K_m}}$ as our regularize, we can obtain sharp learning rates via the proposed method for multi-kernel SVM.
>
> Moreover, we add some experimental results to verify the effectiveness of our method over the state of  art methods for multi-kernel SVM. We use the penalty term of $L_1$-norm as benchmark.  We set $s=0.1$ and $\tau=0.2$, the other settings are the same as the manuscript. The averaged hinge loss on the test data with different size of train data is given in the following table. We can see that the line of best fit of  our mix-norm sparse MKL for empirical excess risks is $sn^{-1/(1 + 1.03\tau)}$ (match the learning rate $sn^{-1/(1 + \tau)}$, Corollary 1)  is faster than that of benchmark method of order $sn^{-1/(1 + 2.02\tau)}$, which demonstrates the effectiveness of our mix-norm based sparse MKL.
>
> | |Method | 200| 400| 600 | 800 | 1000|1200|Learning rate|
> |:-------:|:--------:|:--------:|:-------:|:-------:|:-------:|:-------:|:-------:|:-------:|
> |L1 Sparse MKL| Empirical Excess Risk |0.0466   | 0.0316|0.0285 |0.0259 |0.0238  |0.0195||
> |L1 Sparse MKL |  Fit Line                    | 0.0417 |0.0288  |0.0262| 0.0253| 0.0219 |0.0202| $sn^{-1/(1 + 2.02\tau)}$|
> |Mix-norm Sparse MKL |   Empirical Excess Risk |0.0433  | 0.0241 |0.0236|0.0229| 0.0216 |0.0183 ||
> |Mix-norm Sparse MKL |  Fit Line | 0.0415 |0.0270 |0.0223| 0.0222| 0.0219|0.0186|  $sn^{-1/(1 + 1.03\tau)}$|
>
> 2. Thanks for your valuable comments and suggestions. In the final manuscript, we will add some statements in term of numerical algorithms of our proposed learning method. Specially, we explicitly give the main procedure of forming the group Lasso-type optimization, which has been studied well on its theory and computational complexity in F. Bach, JMLR, (2008). Besides, we add a table to take detailed comparisons with existing related work as follows:
>
> | Method  | Penalty| Optimization| Convergence rate|
> |:-------:|:--------:|:--------:|:--------:|
> |Cortes et.al(2010)|   $\sum_{m=1}^M\|f_m\|_{K_m}$                                 | Group Lasso  | $\log (M)/\sqrt{n}$ |
> |Raskutti et.al(2012) | $\sum_{m=1}^M\|f_m\|\_{n}+\gamma_n\sum_{m=1}^M\|f_m\|_{K_m}$ | SOCP         | $sn^{-\frac{1}{1+\tau}}+\frac{s\log M}{n} $|
> |Zhao and Liu(2012)   | $\sum_{m=1}^M\|f_m\|_{2}$                                   | Proximal GD  | $s\sqrt{\frac{\log M}{n^{4/5}}}$          |
> |This Paper|           $\sum_{m=1}^M\sqrt{\|f_m\|\_{n}^2+\gamma_n\|f_m\|^2_{K_m}}$  | Group Lasso  | $\Big(sn^{-\frac{1}{1+\tau}}+\frac{s\log M}{n}\Big)^{\frac{\kappa}{2\kappa-1}} $ |
>
> Additional numerical experiments on real world datasets (Delve / splice and UCI / mushrooms) shows that our method  give compared time as $L_1$-norm method.
>
> | Method| Accuracy /Time | mushrooms| splice|
> |:-------:|:--------:|:--------:|:--------:|
> |L1 Sparse MKL| Accuracy |0.973| 0.887|
> |L1 Sparse MKL| Time |  11.212(s) |  1.547(s) |
> |Mix-norm Sparse MKL |Accuracy | 0.985|0.896|.
> |Mix-norm Sparse MKL| Time | 11.824(s)  |  1.571(s) |
>
> 3. It is worth noting that, although our method can be reduced to the group lasso optimization like the standard MKL, our proposed SVM involves two hyperparameters $(\lambda_n,\gamma_n)$, resulting in a higher computational cost when choosing them via cross validation.  In practice, if the standard MKL with penalty $\sum_{m=1}^M\|f_m\|_K$ generates too sparse solutions with a greater bias, we advise to apply for our proposed method as an alternative one, due to the simple fact that the empirical norm $\|\cdot\|_n$ is a milder sparsity-induced term than  $\|\cdot\|_K$.
>
> 4. Thank you very much for pointing out some typos / mistakes. We must take them seriously and revise them.

---

> > ### Comment · Reviewer_xwyd · 2021-08-11
> > **Convincing answers to my remarks**
> >
> > Thank you for the details answers that helped clarify points I think are important.
> > To be honest, I don't really believe the "experiments on real world datasets" are adding that much value to an already strong contribution so I would not mind not seeing it in the final version of the paper.
> > Otherwise, I think the more details given in this answer (or to other reviewers) about the computational aspects (how to solve the suggested formulation, empirical convergence rates for various formulations...) are really useful, especially for readers that are not very familiar with the literature and for whom it probably won't be too obvious, for instance, that already existing and efficient group lasso algorithms directly apply.
> >
> > So to be completely clear, are you intending to add that extra material (the tables, elements of discussion) into the manuscript? Do you manage to make it fit in?
> > I am not sure that I am going to increase my already fairly high rating. But I am even more convinced that this submission should be accepted.

---

> > > ### Author Response · Authors · 2021-08-11
> > > **Thank you for your appreciation and suggestions of this manuscript**
> > >
> > > Thanks for your recognition of this paper and useful suggestions. To be sure, we will add that extra material (the tables, elements of discussion) into the manuscript (If we cannot fit in the main body, I will add them in the appendix). Thank you again.

---

### Official Review · Reviewer_k7x5 · 2021-07-28

**Rating:** 6
**Confidence:** 3

**Summary:**

This paper derives generalization bounds for the sparse multi-kernel Support Vector Machine (SVM) problem. Recall that previous literature has focused on sparse multi-kernel problems, but only for the Ridge regression (square loss instead of hinge loss), or on multi-kernel SVM but without the sparsity inducing penalization. To do so, the authors introduce a new penalization, see Equation (5). Under some assumptions already used in previous works, excess risk bounds and estimation error bounds are derived (Theorem 1), further explicited in terms of the eigendecay of the kernel (Corollary 1). Bounds on the classification error are also provided (Theorem 2).

**Limitations And Societal Impact:**

Yes

**Main Review:**

Although I do not question the technical quality of the present submission, I feel that authors fail at conveying a clear message that would benefit the Neurips audience. Among the main criticisms:

- the scope of the paper appears to be relatively restricted, as only the hinge loss is studied
- I might have missed something, but I cannot see why penalizing $||f_m||_n$ makes $\hat{f}$ sparse. As far as I understand, the proposed penalization is some kind of mixed norm (up to the regularization parameters $\lambda_m$ and $\gamma_m$), whose outer norm is $||\cdot||_1$ and inner norm is $\sqrt{||\cdot||_n^2 + ||\cdot||_K^2}$. By the $\ell_1$ outer norm, the solution to (5) contains many components $f_m$ with null inner norm, which means $||f_m||_n^2 + ||f_m||_K^2 = 0$, or again $f_m = 0$ as $||f_m||_K^2 = 0$. But I cannot see where is the impact of $||\cdot||_n$, as we can have $||f||_n = 0$ but $f \ne 0$. Am I missing something here?
- moreover, problem (5) is solved with penalization AND norm constraints (the optimization is carried out on $B_M$). Usually one of the two is chosen but not both at the same time. Is there a particular reason? Overall, I feel this penalization could have been better introduced and motivated, as it is far from natural from the current reading
- another point which is not discussed at all by the authors is the optimization of criterion (5). I think this aspect is very relevant for the Neurips community, and should not be omitted. A better theoretical objective which is impossible to minimize in practice would attract less interest. On the contrary, if the optimization of (5) is simpler than previous algorithms, this is definitely something that should be mentioned
- the bounds feature constants that are not explicited (e.g., $C_0, C_1$), making them harder to interpret. In addition, criterion (5) involves a lot of hyperparameters ($\lambda_m, \gamma_m$), to the extent that it is difficult to extract a clear and direct message from the Theorems, nor to determine how the new penalization was key. The comparison in Section 3.3 is however useful.
- the experiments are very basic and do not present any benchmark

Minor comments:
- l. 12: has bec*o*me
- l. 24/32: the acronyms DC and RKHS are never defined
- throughout the paper, authors keep referring at future equations, sections, which makes the reading incomfortable
- l. 43: not clear around citation [46]
- l. 100-101: RKHSs *are* known
- l. 156: we *knew*
- l. 200: increase*s*
- l. 254: *in this paper* twice
- l. 323: improve on previous

Overall, I think the present contribution is below the NeurIPS acceptance bar: better motivation/intuition/interpretation of the results seem needed, as well as a more in depth study of the algorithmic and empirical aspects.

**Post-rebuttal edit**

I thank the authors for their response, which addressed most of my concerns. Incorporating these comments into the submission (to the extent it is possible with the page limit) would be helpful I think. I won't oppose acceptance and have increased my score to 6 accordingly.

**Time Spent Reviewing:**

5

---

> ### Author Response · Authors · 2021-08-09
> **Response to Reviewer k7x5**
>
> Thanks for your comments, and you pointed out the deficiencies of our paper and the careless of us. According to your comments, we will make clear statements in the final version as follows:
> 1. In view of the popularity of SVM in machine learning, this paper focuses on theoretical investigation on the hinge loss with a mixed functional norm under multi-kernel setting. In fact, the current technical analysis can be easily extended to any Lipschitz loss case, e.g., the Huber loss and the quantile loss used for robust methods.
>
> 2. In general, we know that $\|\hat{f}\|_n=0$ does not imply $f=0$. However, in the case of  any kernel-based minimization problem, we can show that $\|\hat{f}\|_n=0$ always implies $f=0$. Our proof is mainly based on the reproducing property of Mercer kernel, $f(x)=\langle f, K_x \rangle_K$ for any $f \in H_K$, where $H_K$ is a reproducing kernel Hilbert space. In fact, if $\|\hat f\|_n=0$ holds, that is, $\hat{f}(x_i)=0$ for all $i=1,\ldots,n$, by the reproducing property we have $\langle \hat f, K\_{x_i} \rangle_K=0\$ for all $i$. Hence, $\hat f$ is orthogonal to the subspace $S_n:=span\\{K\_{x_1},...,K\_{x_n}\\}$. On the other hand, using the reproducing property again, any solution $\hat f$ of kernel-based minimization problems  has a finite representation within $S_n$. So we conclude $\hat{f} =0$. Although several previous work (e.g. M. Kloft, U. Brefeld, S. Sonnenburg, and A. Zien., JMLR, 2011; F.Bach, JMLR,2008) have used the kernel norm $\|f_m\|_K$ as a sparsity-induced regularization, sharp learning rates cannot be given under standard conditions, mainly because a weight empirical process inequality involves the empirical norm $\|\cdot\|_n$, which is very crucial to derive better learning rates. Similar theoretical studies have been well-established for least square loss cases, see Suzuki and Sugiyama. AIStat, 2012; Raskutti,  Wainwright, and  Yu, JMLR,2012; Meier, de Geer, and  Bühlmann, AOS, 2009. Note also that $\|f\|_n\leq \kappa \|f\|\_K$, where $\kappa:=\sup\_{x}|K(x,x)|$. This implies that  $\|\cdot\|_n$ has a weaker constraint than $\|\cdot\|_K$, so that the solution of $\hat f$ is more flexible when $\|\cdot\|_n$ is used to generate sparse components.
>
> 3. Thanks for your detailed comments. Usually, a regularized learning scheme is equivalent to a norm-constraint optimization problem. One of the two has been chosen in most  existing related work. However, we are also interested in high dimension problems that have received great attention in statistical literature, where the number of base kernels is allowed to diverge as $n\rightarrow \infty$. In this case, the norm-constraint $B_M$ is often imposed for only technical purpose. See a similar formulation in the learning scheme considered in Raskutti,  Wainwright, and  Yu, JMLR, 2012. The use of our mixed norm regularization is mainly motivated by the following fact: i) our proposed estimation with the mixed norm can lead to very sharp learning rates; ii) the empirical norm $\|\cdot\|_n$ used for sparsity is much milder than $\|\cdot\|_K$.
>
> 4. By finite representation of reproducing kernel, each additive estimator $f_m(\cdot)=\sum_{i=1}^n\alpha_i^mK_m(x_i,\cdot)$ for all $m=1,...,M$. A direct computation leads to
> $$
> 	\sqrt{\|f_m\|_n^2+\gamma_n\|f_m\|^2_K}=\sqrt{(\alpha^m)^T\widetilde{K}_m\alpha^m}
> $$
> where $\widetilde{K}_m=\frac{\mathbb{K}_m^2}{n}+\gamma_n \mathbb{K}_m$. Here $\mathbb{K}_m$ is the kernel matrix induced by $K_m$ at points $(x_i)$. Note that $\widetilde{K}_m$ is a semi-definite matrix, which can be written as $\widetilde{K}_m=A^2$ with some matrix $A$. Hence, our original learning scheme in (5) is transformed into a group Lasso optimization, and there exist several efferent numerical algorithms for solving it, such as proximal methods and coordinate descent ones.
>
> 5. $(C_0,C_1)$ are two constants independent of $n, M$ or $s$. Their definitions rely on the result of  and Proposition 5 in Koltchinskii and Yuan AOS, (2010), where their constant did not give an explicit form. So, we can not give a more explicit form on $(C_0,C_1)$. Since $\gamma_n=\lambda_n^2$ in our theory,  there is no additional hyperparameter to be optimized. To explain the role of two hyperparameters, we rewrite the mixed penaltation with two different parameters as:
> $$\lambda_n\sqrt{\|f_m\|_n^2+\gamma_n\|f_m\|^2_K}=\sqrt{\beta_n\|f_m\|_n^2+\theta_n\|f_m\|^2_K}.$$
> We see from the above equation that, $\beta_n$ is used to control sparsity, while $\theta_n$ is used to control functional smoothness, due to the fact that $\theta_n$ is a smaller order of $\beta_n$, precisly, $\theta_n=\beta_n^2$.
>
> 6. Thanks for your valuable suggestions. To verify the effectiveness of our mix-norm based sparse MKL, some experiments on both synthetic datasets and real world datasets are added. We use the popular $L_1$-norm sparse MKL as benchmark.
>
> All experiments are implement on Google's TensorFlow framework. On synthetic datasets, we set $s=0.1$, $\tau=0.2$, the other settings are the same as the manuscript. The averaged hinge loss on the test data with different size of train data is given in the following table.  We can see that the line of best fit of our mix-norm sparse MKL for empirical excess risks is $sn^{-1/(1 + 1.03\tau)}$ (match the learning rate $sn^{-\frac{1}{1 + \tau}}$, Corollary 1)  is faster than that of benchmark method of order $sn^{-1/(1 + 2.02\tau)}$, which demonstrates the effectiveness of our mix-norm based sparse MKL.
>
> | |Method | 200| 400| 600 | 800 | 1000|1200|Learning rate|
> |:-------:|:--------:|:--------:|:-------:|:-------:|:-------:|:-------:|:-------:|:-------:|
> |L1 Sparse MKL| Empirical Excess Risk |0.0466   | 0.0316|0.0285 |0.0259 |0.0238  |0.0195||
> |L1 Sparse MKL |  Fit Line                    | 0.0417 |0.0288  |0.0262| 0.0253| 0.0219 |0.0202| $sn^{-1/(1 + 2.02\tau)}$|
> |Mix-norm Sparse MKL |   Empirical Excess Risk |0.0433  | 0.0241 |0.0236|0.0229| 0.0216 |0.0183 ||
> |Mix-norm Sparse MKL |  Fit Line | 0.0415 |0.0270 |0.0223| 0.0222| 0.0219|0.0186|  $sn^{-1/(1 + 1.03\tau)}$|
>
>
> We also add some experiments on real world datasets. We use two publicly available datasets, mushrooms and splice, from the LIBSVM Data. For mushrooms dataset, we first split it into train set (70%) and test set (30%). The splice dataset already been split, so we keep the original settings, the train set has 1,000 records and the test set has 2,175 records.
> The accuracies of L1 sparse MKL and our mix-norm sparse MKL are given in the following table. We can also find that our mix-norm sparse MKL is effective.
>
> | Method| Accuracy /Time | mushrooms| splice|
> |:-------:|:--------:|:--------:|:--------:|
> |L1 Sparse MKL| Accuracy |0.973| 0.887|
> |L1 Sparse MKL| Time |  11.212(s) |  1.547(s) |
> |Mix-norm Sparse MKL |Accuracy | 0.985|0.896|.
> |Mix-norm Sparse MKL| Time | 11.824(s)  |  1.571(s) |
>
>
> 7. Thank you very much for pointing out some typos / mistakes. We must take them seriously and revise them.

---

> > ### Comment · Reviewer_k7x5 · 2021-08-30
> > **Post-rebuttal edit**
> >
> > I thank the authors for their response, which addressed most of my concerns. Incorporating these comments into the submission (to the extent it is possible with the page limit) would be helpful I think. I won't oppose acceptance and have increased my score to 6 accordingly.

---

> > > ### Author Response · Authors · 2021-08-30
> > > **Thanks for the feedback**
> > >
> > > Thank you very much for the appreciation of our response and the update on the score. We will incorporate your valuable comments into the final version. Thank you again.
> > >
> > > Best, Authors

---

### Author Response · Authors · 2021-08-09
**Thanks for the four anonymous reviewers**

We would like to thank four anonymous reviewers for their valuable comments and rapid responses, and their useful comments and suggestions help us improve the manuscript greatly. We attempt to answer all the questions involved in the review report. The following is a point-to-point response to the four reviewers.

---

### Author Response · Authors · 2021-08-28
**Dear AC and Reviewers**

Thank you again for the great effects and valuable comments. We have carefully addressed the main concerns in details. We hope you might find the response satisfactory. As the discussion phase is about to close, we are very much looking forward to hearing from you about any further feedback. We will be very happy to clarify any further concers (if any).

Best, Authors

---

### Decision · Program_Chairs · 2021-09-27

**Decision:**

Accept (Spotlight)

**Comment:**

All reviewers have praised the quality of the paper (both substance and form) and acknowledged their appreciation of the responses made by the authors to their comments. This is a clear accept, with an expectation from the authors to do their best to incorporate as much as the insightful comments they provided in the feedback into the main text.